# Water Pathways for the Hindu-Kush-Himalaya and an Analysis of Three Flood Events

**Robert Boschi [1,*] and Valerio Lucarini [1,2,3]** 

[1]  CEN, Meteorological Institute, University of Hamburg, 20144 Hamburg, Germany
[2]  Department of Mathematics and Statistics, University of Reading, Reading RG6 6AX , UK
[3]  Centre for the Mathematics of Planet Earth, University of Reading, Reading RG6 6AX, UK
[*]  Correspondence: robertboschi@gmail.com

**Abstract:** The climatology of major sources and pathways of moisture for three locales along the Hindu-Kush-Himalayan region are examined, by use of Lagrangian methods applied to the ERA-Interim dataset, over the period from 1980 to 2016 for both summer (JJA) and winter (NDJ) periods. We also investigate the major flooding events of 2010, 2013, and 2017 in Pakistan, Uttarakhand, and Kathmandu, respectively, and analyse a subset of the climatology associated with the 20 most significant rainfall events over each region of interest. A comparison is made between the climatology and extreme events, in the three regions of interest, during the summer monsoon period. For Northern Pakistan and Uttarakhand, the Indus basin plays the largest role in moisture uptake. Moisture is also gathered from Eastern Europe and Russia. Extreme events display an increased influence of sub-tropical weather systems, which manifest themselves through low-level moisture transport; predominantly from the Arabian sea and along the Gangetic plain. In the Kathmandu region, it is found that the major moisture sources come from the Gangetic plain, Arabian Sea, Red Sea, Bay of Bengal, and the Indus basin. In this case, extreme event pathways largely match those of the climatology, although an increased number of parcels originate from the western end of the Gangetic plain. These results provide insights into the rather significant influence of mid-latitudinal weather systems, even during the monsoon season, in defining the climatology of the Hindu-Kush-Himalaya region, as well as how extreme precipitation events in this region represent atypical moisture pathways. We propose a detailed investigation of how such water pathways are represented in climate models for the present climate conditions and in future climate scenarios, as this may be extremely relevant for understanding the impacts of climate change on the cryosphere and hydrosphere of the region.

**Keywords:** monsoon; precipitation; climate; weather extremes; moisture pathways; Lagrangian modeling Hindu Kush Himalaya; Pakistan; Uttarakhand; Kathmandu

## 1. Introduction

The mountainous regions of the Hindu-Kush-Karakoram-Himalaya (*HKKH*) contain nearly 4000 km$^3$ of ice, which is the third largest mass of ice on Earth, with only Antarctica and the Arctic/Greenland supporting more. However, unlike the polar regions, the *HKKH* directly supports the livelihoods of 200 million people and provides water and ecosystem services to 1.3 billion people, a fifth of the global population, making up parts of Afghanistan, Bangladesh, Bhutan, China, India, Nepal, Myanmar, and Pakistan through ten large Asian river systems: the Amu Darya, Indus, Ganges, Brahmaputra (Yarlungtsanpo), Irrawaddy, Salween (Nu), Mekong (Lancang), Yangtse (Jinsha), Yellow River (Huanghe), and Tarim (Dayan) (see Figure 1). Thus, changes driven by unprecedented climate change in the moisture supply to the *HKKH* region, which maintains the glaciers and provides fresh



snow cover, is a major source of concern. Through rainfall and snow-melt, agriculture and power generation are heavily dependent on reliable sources of water. For the Indus Basin, the first available source of water after the dry season, from October to March, comes from snow-melt of the *HKKH* [1]. As far as the Indus, Gange, Brahmaputra, and Mekong are concerned, the basins have become wetter due to more intense monsoonal precipitation, and the contributions of snow- and ice-melt have become less relevant [2]. An unexpected consequence to changes in the *HKKH* water supply has been highlighted in recent studies by [3,4], where it was proposed that changes in the Earth's axis are related to water mass loss away from the Indian subcontinent and the Caspian Sea.

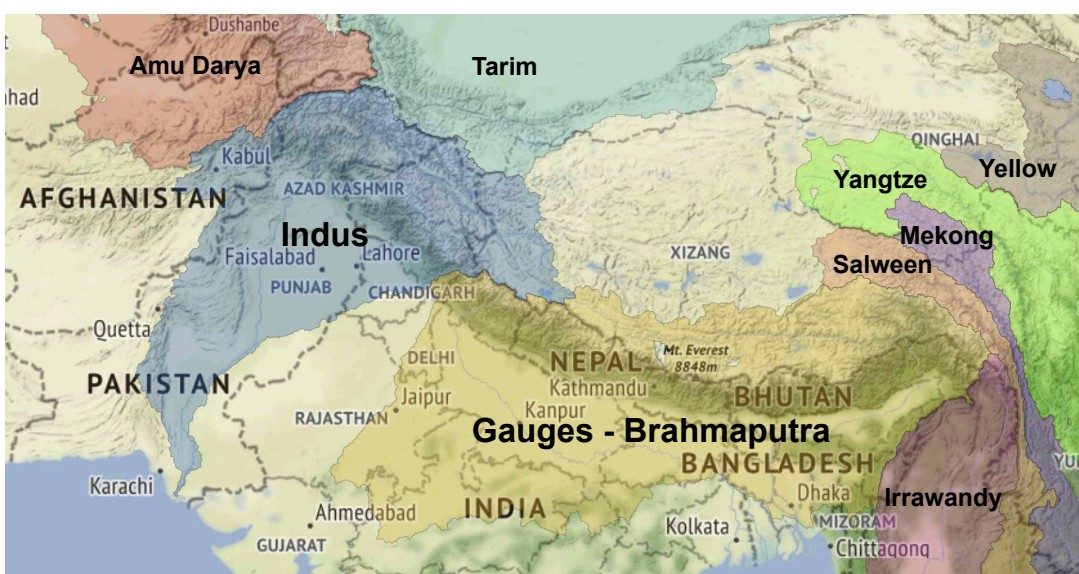

**Figure 1.** The ten main river basins located in the Hindu-Kush-Karakoram-Himalaya (*HKKH*) region (from west to east): Amu Darya, Indus, Tarim, Ganges, Brahmaputra, Irrawaddy, Salween, Mekong, Yangtze, and Yellow River basins.

The complex and irregular orography of the *HKKH* region means that accurately modelling and taking sufficient observations to survey the region adequately is rather difficult from ground level. An analysis by [2,5] showed that the PCMDI, CMIP3, and the more recent CMIP5 models have many issues in reproducing the observed climate conditions of the *HKKH* region—in particular, during the monsoon—including for the accurate representation of the timing of its onset and decay. Some of these issues pertain to an inadequate representation of the boundary conditions, such as those associated with horizontal resolution limitations, which lead to an overly smoothed orography west of the Tibetan Plateau, resulting in an unrealistic intrusion of cold air which suppresses moisture convection and reduces monsoon precipitation. As a consequence, the lower level jet subsides, preventing the full west and northwest extent of the monsoon, ultimately delaying its onset. A poor performance in simulating the monsoonal state was, thus, reflected in the model's ability to correctly represent river runoffs. An analysis by [5] also showed that PCMDI/CMIP3 models do not conserve water at the river basin scale and are neither consistent with observation nor other models, with most having a tendency to underestimate the net water budget . for the four considered river basins (Indus, Ganges, Brahmaputra, and Mekong). Furthermore, a lack of irrigation parametrisations within the models makes it impossible to correctly represent the local hydrological cycle in regions where artificial irrigation is widespread, as in parts of the Indus and Ganges; see the discussion in [6–8]. Overall, the model performance decreases from the east to the west and northwest, meaning the Indus and Ganges basins are not sufficiently well-represented by present-day Earth system models, which has significant implications for assessing the impact of climate change.

From 1990 to 2001, the entirety of the Himalayas witnessed a decrease of approximately 16% in snow cover [9]. This has continued to be the case, where [1] has shown that, from 2000 to 2008,

a similar trend has continued. The complex orography, coupled with the fact that this region sits in an area influenced by both tropical and mid-latitudinal weather systems, means that the local and seasonal picture is complicated, displaying a varied response to changes in the climatic state. Recent studies have highlighted that the northwestern Himalaya have shown less glacier shrinkage than the eastern parts of the mountain range ([10–12]).

Conversely, the west and central Karakoram regions have exhibited irregular behaviour, with observations indicating a possible mass increase since the 2000s [11–15]. As an example, the upper Shyok valley has exhibited heterogeneous behaviour, with an overall increase from 1973 (1613.6 $\pm$ 43.6 km$^2$) to 2011 (1615.8 $\pm$ 35.5 km$^2$), having initially decreased between 1973–1989 [16]. It has been noted that 75% of the area change over this period could be attributed to surge-type glaciers, while the remaining glaciers remained largely stable. The average size of glaciers which tend to exhibit surge-type behaviour is rather large (56 km$^2$), compared to non-surge type glaciers (approximately 1 km$^2$). Since 1989, the number of surge-type glaciers has increased (from three to 18) in the upper Shyok valley. As well as the effect on the overall glacier size, the environmental impacts of surge glaciers are felt by the communities downstream, as the result of blockages which lead to the formation of lakes that periodically burst their banks, releasing large amounts of debris-filled water. Additionally, 18 of the 136 glaciers examined in detail showed no significant sign of change during this period.

The investigation reported in [17] showed that glaciers behave differently in the central Himalaya and in the Karakoram region, and that the glacier mass balance is sensitive to both the residual flux of the energy balance and snowfall, implying that the sensitivity of glaciers to climate change is regionally driven. In the Karakoram, there is an additional sensitivity to summer precipitation by means of an albedo increase and summer cloudiness. Their results implied that, in the event of there being increased precipitation frequency due to a stronger summer Indian monsoon, the mass balance of Karakoram glaciers would increase, relative to the Central Himalaya.

In addition to the socio-economic issues to habitability caused by long-term climatological changes, extreme hydrological events, such as floods and droughts, which occur due to large excesses or deficiencies in localised precipitation, often lead to a massive loss of lives as well as livelihoods; consider, also, the non-trivial impacts of changes in the hydroclimate (and, especially, in its extreme events) of mountainous regions on the advent of landslides [18,19]. Such recent devastating events include the 2010, 2013, and 2017 floods of Northern Pakistan (*NP*), Uttarakhand (*U*) and Kathmandu (*K*), respectively. As an example, the 2010 flood submerged roughly 20% of Pakistan's land area, killing and injuring nearly 5000 people directly, while 20 million more were displaced. In this paper, we will investigate what distinguished these events from the climatology.

While anomalous snow-melt, water management practices, and years of drought (leading to reduced vegetation cover and reduced retention capacity of the soil) all contributed to the catastrophic flood scenario that unfolded [20] in *NP*, the main cause was a series of monsoonal deluges during July 2010, which coincided with a persistent blocking event which began in June and lasted about 2 months. Relative to the usual climatology, this high pressure system occurred further east over Russia than usual, which meant that the trough to its east extended southward to *NP*. The effect of this was to bring high potential vorticity (PV) air to the region, which drove convection and low level convergence, drawing moist monsoonal air from the Indian subcontinent and the Arabian Sea further north and west than usual [20–27]. The fact that El Niño-Southern Oscillation (ENSO) was in a La Niña phase may have also led to stronger westerly winds over the Bay of Bengal and the Indian subcontinent. Analysis of the *U* flood in June 2013 showed that a similar mechanism caused the heavy precipitation events seen between the 14–17 June [28–35]. The north-westward extent of the monsoon was much further along, compared to normal, in mid June. At the same time, between 12–17 June, a deep trough developed with a westward tilt over the region before fading and weakening of the easterly jet. The August 2017 flood event that impacted Kathmandu and Nepal as a whole, with heavy impact on the southern region of Terai, has been studied less extensively from a meteorological point of view. It appears that tropical–extratropical dynamics play a key role in causing extreme events in the

*HKKH* region. During the summer months, there exists a circumglobal teleconnection pattern with a zonal wavenumber of 5, which represents the second leading empirical orthogonal function (EOF) of interannual variability of the upper tropospheric circulation and is associated with the westerly jet stream. The investigation of [36] found that this is correlated with precipitation events in regions which include the Hindu-Kush, implying that there is a link between specific configurations of the westerly jet stream with the occurrence of precipitation over the Hindu-Kush region during the monsoon season.

The distance over which a precipitation event gathers moisture will, of course, vary widely for individual systems and different synoptic environments, depending on whether these are small- or large-scale systems, on average. [37] suggested that the moisture supply for rainfall-producing weather events locally relies heavily on advective sources of moisture, originating globally from distances of about 3–5 times the radius of the precipitation region. From this perspective, local precipitation greatly depends on the transport of moisture from other regions by the atmosphere. Thus, in most cases, a geographic, local focus on precipitation extremes provides a rather limited diagnostic perspective, without the ability to properly study the underlying causes leading to precipitation events. It is, thus, far more useful to consider the sources and variability of the advected moisture, in order to understand the causes of hydrological extremes [38,39]. Furthermore, by analysing the variability in moisture source regions (e.g., strength and distribution) in the context of long-term climate change, the impact on moisture availability and subsequent precipitation variability over target regions can be better understood.

Unlike Eulerian numerical methods, Lagrangian parcel analysis allows the observer to directly follow the evolution of a parcel's properties along its trajectory (see [40,41] for an extensive overview of Eulerian and Lagrangian analysis of the water budget and transport at a global level, respectively). In this case, it is therefore possible to directly follow an air parcel from a given location backwards in time, from its final destination to its origin, while also tracking changes to its moisture content, temperature, and so on. We have been inspired, here, by the analyses presented in such studies as [24,42–44]. For instance, in 2002, there was a flood event over Dresden [42,45,46] where a nearby weather station measured a third of the annual precipitation to fall over a 24 h period between 12–13 August. This anomalous precipitation has been attributed to a Genoa Vb cyclone crossing the Alps from southwest to northeast. In [42], Lagrangian techniques were used to backtrack the precipitated moisture and describe the spatial pattern of its origin.

The purpose of this paper is to complement previous analyses based mainly on Eulerian methods (see e.g., [23,47]) and consider the moisture sources for precipitation occurring over the NP, U, and K regions over a variety of time scales and classes of precipitation events. Using a customised (see the next section for details) version of the Flexible Particle dispersion model (FLEXPART) which has been developed and validated by [42,43] to analyse moisture transport, we construct the climatology of the moisture sources for precipitation during the summer and winter seasons over the 1980–2016 period. In addition, we consider specific case studies of the aforementioned extreme flood events of the 2010s in the *NP, U*, and *K* regions. Finally, we analyse the 20 strongest summer precipitation events which occurred during 1980–2016 over these regions. In doing so, we aim at reaching a better understanding of the moisture transport mechanisms affecting these regions. In particular, this analysis aims to show whether there are discernible differences in the moisture paths and sources which lead to extreme events in these regions and, therefore, obtain a better understanding of the conditions which led to these events. In other terms, we would like to ascertain whether the extreme events were, in some sense, just intensified typical events, or whether there was something special, dynamically speaking, about them. Our study is performed on ECMWF re-analysis data, but it would be of clear interest to test to what extent the IPCC-class climate models are able to replicate the structure and prevalence of the water pathways, and how they project future changes. To this end, the paper will be structured as follows: In Section 2, we describe the data and methods used in this study. In Section 3, we discuss the climatology of the water transport for the three study locations, describing how the characteristic patterns of the moisture sources vary from west to east for the summer and winter periods. Section 4

is focused on making comparisons between the summer climatology and the three case studies of recent extreme flood events to have effected *NP*, *U*, and *K*. Furthermore, we analyse a subset of the climatology containing the 20 most significant extreme precipitation events to hit each of our regions of interest and compare the differences in the moisture pathways and sources to the overall climatology and the case studies. Finally, we summarise the discussion in Section 5.

## 2. Data, Methods, and Model Description

### 2.1. Data Sources

#### 2.1.1. APHRODITE

APHRODITE [48] is a daily, 0.25° resolution, gridded, and gauge-based precipitation product which covers the period from 1951–2007. It is based on data collected from 5000–12,000 stations and performs relatively well against satellite-only or blended products, especially over orography [49].

#### 2.1.2. ECMWF

ERA-Interim [50] is a global, 0.7° re-analysis product developed by the European Centre for Medium-Range Weather Forecasting (ECMWF). It has 60 vertical model levels from the surface to 0.1 hPa, 27 of which are below 100 hPa, and covers a period from 1979 to the present day.

### 2.2. Model Description and Methods

The Lagrangian atmospheric transport model, FLEXPART, uses operational data with $1° \times 1°$ resolution from the European Centre for Medium Range Weather Forecasts (ECMWF) interim re-analysis [50]. The model, using the three-dimensional wind fields of the ECMWF analyses, transports atmospheric parcels backward in time. Parcel positions and properties are calculated and stored. For the Lagrangian trajectories [43], the ECMWF global analysis dataset was used every 6 h (at 0000, 0600, 1200, and 1800 UTC) and 3 h forecast data at intermediate times (at 0300, 0900, 1500, and 2100 UTC) on the 60 model levels over a period of 36 years, between 1980–2016.

FLEXPART was started on 31 December 2016 and, for each considered day, approximately 1,000,000 parcels were generated, from the Earth's surface to 10,000 m above the ground, over the Hindu-Kush-Himalaya (*HKH*) and then moved backward, according to the advection described by the ECMWF wind fields, for 10 days. Although the simulation period was relatively short and may miss low-frequency variations in the climate, it was sufficient to reveal the key characteristics of moisture flux and the pathways and source regions associated with extreme moisture events and climate. Clearly, as we delve into regions featuring extreme orographic features, one needs to notice that the ECMWF resolution might be a bit coarse and lead to imprecision in our results. Indeed, an implementation of FLEXPART which is able to interface with limited area models has been presented in the literature (see, e.g., [51,52]). Nonetheless, since we are considering very large-scale transport phenomena, the classic version of FLEXPART in conjunction with ECMWF was adopted for our investigation.

### 2.3. Implementation and Use of a Modified FLEXPART Advection Model: Moisture Sinks and Sources

In what follows, we summarise the key details of the model, including the modifications made, and discuss how we use the FLEXPART output data to determine the moisture sources and sinks necessary for this work. Detailed descriptions of the Lagrangian method to track moisture in the atmosphere, which has been used in numerous studies, can be found in [42–44]. Here, we apply a similar analysis using an updated version of the model containing numerous enhancements.

The model is set up in domain filling mode, where the model divides the atmosphere into N parcels, each having constant mass $mp = ma/N$, where *ma* is the total mass of the atmosphere. By default, the model is not set up to allow back-trajectories to be calculated in domain fill mode and, thus, we needed to remove these limitations in the code before running the model, in order to work

with our experimental setup. As part of our analysis, we also required knowledge of which parcels precipitated over our regions of interest. To this end, we also modified the code to calculate the relative humidity, at time $t = 0$, of the back-trajectories, by interpolating the relative humidity of the ECMWF data to each parcel position. During subsequent time steps, the model, using the three-dimensional wind fields of the ECMWF analyses, transported each parcel in time and the values of the various parcel properties, including the specific humidity ($q$), relative humidity ($RH$), parcel pressure ($p$), and density, were interpolated to the parcels positions again.

Then, from the model output, it is possible to diagnose the moisture sources and sinks for those parcels which were en route to our regions of interest and precipitated there at the target time. This process is essentially performed, for each parcel, as follows: The ECMWF model is parameterised in such a way that parcels precipitate with a relative humidity of roughly 80% [44]. Therefore, taking all air parcels that are precipitating ($RH > 80\%$) within the target region at t = 0, we may isolate the parcel trajectories directly associated with the precipitation event. These parcels can, then, also be used to provide a precipitation estimate of the column for the region of interest,

$$P_{sfc} = -\frac{1}{g} \sum_{k=1}^{k_{top}} \Delta q(t = 0) \cdot 10^{-3} \Delta p_k, \tag{1}$$

where $t = 0$ is the initial time for the back-trajectories over the target region and

$$\Delta q = q(x(t)) - q(x(t - \Delta t)), \tag{2}$$

where $q$ is a parcel's specific humidity at $t = 0$, $\Delta t$ is the temporal resolution for the data used (3 h), $g$ is the acceleration due to gravity, and $\Delta p_k$ is the vertical extent of the parcel for each vertical index $k$. We wish to find the times and locations for each parcel where it received more moisture through evaporation ($E$) than it had precipitated ($P$); that is, where $E - P > 0$ was satisfied. Generally, this is a sufficient criteria to account for changes in moisture dq/dt by an air parcel for a given period of time, such that

$$\frac{Dq}{Dt}\Delta t \approx E(\Delta t) - P(\Delta t) = q(x(t)) - q(x(t - \Delta t)), \tag{3}$$

where, $E_{\Delta t}$ and $P_{\Delta t}$ define the temporally cumulated (over time $\Delta t$) evaporation and precipitation, respectively. Thus, $\Delta q / \Delta t > 0$ indicates the uptake of moisture by a parcel along its trajectory.

In the following, the moisture sources and sinks and their contributions to precipitation in the target region at the target time are considered in several different climatological scenarios. For the climatology, we consider the back-trajectories of all parcels originating from a specific region for all days in the considered range. Overall, this requires us to account for an extremely large amount of data. In the climatology, we account for a given property of the parcel, such as $\Delta q$, as a vertical column integral on a $1 \times 1$ degree grid at each time step, integrated over all time steps ($Q_c$), as follows:

$$Q_c = \frac{1}{g} \sum_{t} \sum_{k=1}^{k_{top}} \Delta q(t = 0) \Delta p_k \Delta t. \tag{4}$$

Thus, $Q_c = (E - P)_t > 0$ and $Q_c = (E - P)_t < 0$ thus correspond to climatological sources and sinks of moisture, respectively, where $(E - P)_t$ indicates the $1° \times 1°$ horizontal grid column integration of $E - P$ at a given time step. This analysis provides us with the tendency for a given region to act as a source or sink of moisture en route to our regions of interest, which shall be denoted by green boxes in subsequent figures. Furthermore, we consider the average particle uptake of each column. Though our interest here is on Pakistan, our climatological analysis also took into account two other

regions along the foothills of the Himalaya, to the East. This was done, in part, to emphasise how the relative influence of mid-latitudinal and sub-tropical weather systems changed from west to east.

In general, over the course of a parcel's life cycle, it will gain and then lose a proportion of its moisture on numerous occasions, through evaporation and precipitation, respectively. Earlier evaporation events will, therefore, contribute less to the main precipitation event of interest, as they will be partially involved in preceding precipitation events before rehydrating again later along its trajectory, as illustrated in Figure 2.

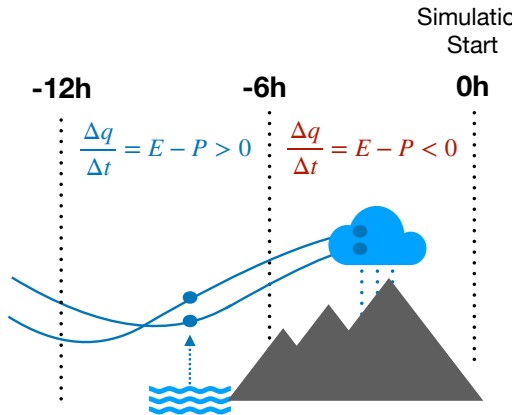

**Figure 2.** Source and sink calculation of the parcel trajectories as individual parcels move backward in time. The dotted arrow indicates a regional source of moisture for a parcel as a result of evaporation (i.e., $E - P$ is positive). The cloud shows parcels losing net moisture (sink) by precipitation (i.e., $E - P$ is negative).

## 3. Climatological Sources and Sinks of Moisture

### 3.1. Summer Season

The sources and sinks of moisture are calculated by considering all parcels originating in our three target regions for the 1980–2016 June–July–August (JJA) season, which roughly correspond to the period when the Indian monsoon is more active. Moving from west to east, these are: Northern Pakistan (*NP*), Uttarakhand (*U*), and Kathmandu (*K*). Note that *K* is in the foothills of Himalaya, but much further to the east along the Gangetic plain, with respect to *NP* and *U*, and further away from the influence of the mid-latitude westerlies. As described above, each parcel's position was accounted for in time and space and both negative and positive moisture uptakes were integrated for each atmospheric column. For benefit to the reader, we provide context to the results which follow in subsequent sections, illustrating the state of the general circulation, between 1980–2016. Figure 3a–c show the seasonal mean winds for the 800 hPa, 500 hPa and 300 hPa pressure levels, respectively, during the summer (JJA). Figure 3d–f show the seasonal mean winds of the 800 hPa, 500 hPa and 300 hPa pressure levels, respectively, during the winter (NDJ).

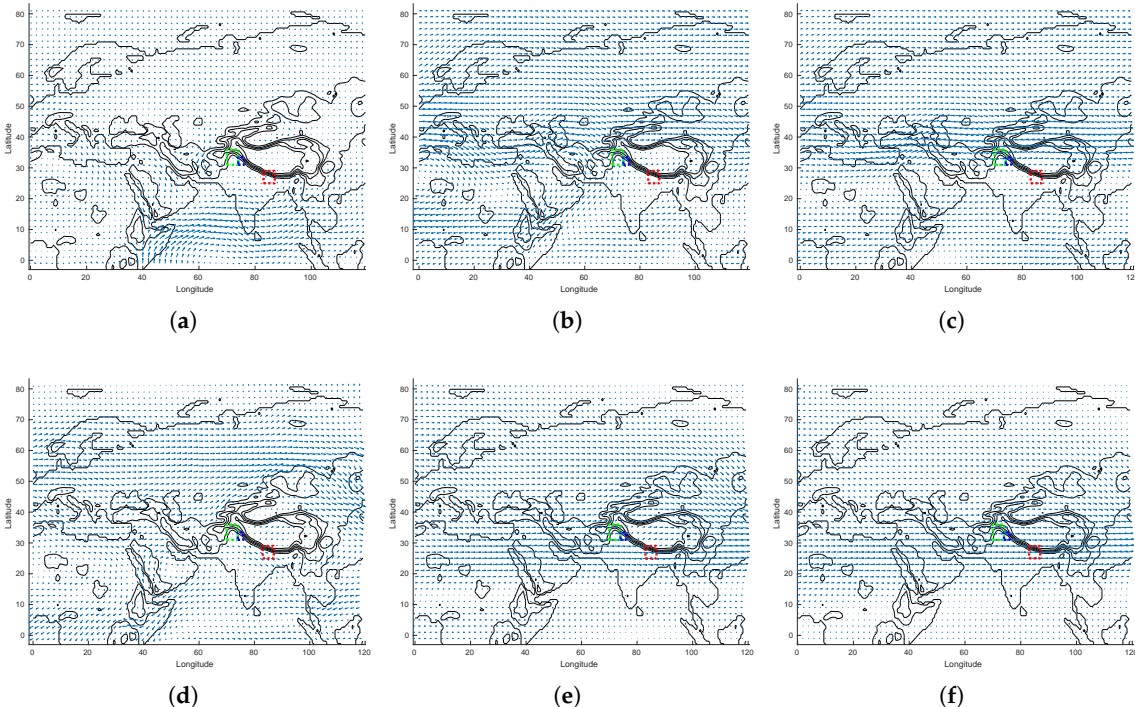

**Figure 3.** Figure showing the seasonal wind fields between 1980–2016 for: (**a**) 800 hPa level for JJA (**b**) 500 hPa level for JJA (**c**) 300 hPa level for JJA (**d**) 800 hPa level for NDJ (**e**) 500 hPa level for NDJ (**f**) 300 hPa level for NDJ. Green, blue and red boxes are *NP*, *U* and *K* regions, respectively.

Figure 4 shows the source and sink contributions, $(E - P)_t$, on $t$ = 2, 6, and 10 days of the back-trajectories in time, for *NP*, *U*, and *K*. Regions characterised by blue colours correspond to $(E - P)_t > 0$ and red ones to $(E - P)_t < 0$. In the former, evaporation dominates over precipitation, which implies that air parcels located within that vertical column gained moisture en route to our target regions (see Figure 2). In the latter, the blue regions were, therefore, considered to be sources and the red regions as sinks of moisture. These results represent temporal averages over 36 years with a temporal resolution of 6 h; although the runs were integrated in FLEXPART using data with a 3 h temporal resolution. It is, therefore, necessary to note that these results represent the climatology for billions of particles summed together, such that each location was a source or sink on average, while individual cases of weather events could be very different. Additionally, a blue (red) source (sink) region indicates that most parcels gained (lost) moisture in that region. In Figure 5, we show the mean height of the parcels coming from a given location: This information is key for understanding the regions where moisture tends to be taken up over. The corresponding parcel densities are also shown in Figure 6, providing a quantitative counterpart to that shown in Figure 4. In addition, we have provided the climatology of the 800 hPa, 500 hPa, and 300 hPa seasonal winds (Figure 6a,d,g), in order to give the reader a context of the state of the general circulation, with respect to our findings.

We will now investigate the main climatological sources and sinks and describe the generalised paths taken by the air parcels en route to each of our regions of interest. This will be done starting in the west and moving eastward, along the Gangetic plain, from *NP* to *K*.

### 3.1.1. Northern Pakistan

For *NP* (Figure 4a,d,g), almost all the air remains close-by over the first two days back in time. Just north of the Hindu-Kush, $(E - P)_2 > 0$ and, so, moisture is picked up. The parcels then passed over the mountain ridge of the Hindu-Kush and were found to mostly precipitate due to orographic lifting, such that $(E - P)_2 < 0$. The target region and the area to the southwest had $(E - P)_2 > 0$,

indicating that these regions produced net evaporation, with water being recycled from the Indus basin, having already precipitated or run down from the Hindu-Kush immediately to the northwest. We consistently see this feature over all time scales, up to the maximum $(E - P)_{10}$ considered here, that a band just north and north-west of the target area mostly precipitates, while another band to the north of this uptakes moisture. We note that our findings indicate that this local moisture uptake is the largest direct moisture source for precipitation over $NP$.

Perhaps somewhat surprisingly, evidence of the the contributions from the Arabian sea immediately to the south-west of Pakistan were non-existent in the climatology. Instead, Figure 6d,g, which show the parcel densities 2 and 6 days back along the trajectories, indicate that the air flow to $NP$ was dominated by air parcels travelling low in the troposphere and originating from Eastern Europe and Russia, thus illustrating the importance of the role played by mid-latitudinal weather systems in bringing air and moisture to the region. Figure 6 also indicates the main path for parcels originating in Eastern Europe and Russia en route to $NP$, which tended to travel at low levels in the troposphere, further illustrating these were moisture source regions. From Figure 4d,g, which shows the moisture sources and sinks, we can see directly that moisture was indeed taken up in those regions.

The region southeast of $NP$ had $(E - P)_6 < 0$, indicating that parcels crossing these regions mostly precipitated water on their way to the target region and did not normally contribute significantly to the precipitation. This conclusion is further supported by the fact that Figure 5 shows that the mean height of parcels en route to $NP$ appeared to be in the mid to upper troposphere. Along the coast of Somalia, there was a region where moisture was collected and, then, carried across the Arabian Sea by the Somalian jet to the Indian subcontinent. Figures 4–6 provide evidence for this phenomenon for all target locations considered here. In the case of parcels moving to $NP$, the parcels which originated along Somalian coastal regions were small in number, and tended to gain altitude and precipitate prior to reaching the target destination. We also note that parcels were also found to travel from the Bay of Bengal along the Gangetic plain, though these parcels also tended to precipitate in the troposphere before reaching $NP$. This is, indeed, consistent with our understanding of how monsoonal circulation works, with $NP$ being its west-most point of reach. Note, also, that no air parcels reached $NP$ from Tibet and China, as they were blocked by the Himalayan range.

The findings above fit with the general knowledge we have of the relatively low precipitation regime received by Pakistan over the course of the year, compared to the rest of the Indian subcontinent, as the air flow dominating the Pakistan region during the rainy season originates from relatively dry air parcels to the North, while most of the moisture contained in parcels from the subtropics is lost en route. Taking a dividing line between the tropics and mid-latitudes at $36°$, we estimate that 45% of parcels originated from the latter (or higher latitudes), 10 days back in time. This indicates the relevance of dynamical links between the mid-latitude and polar regions and the Asian subcontinent. Furthermore, low-altitude air parcels travelling to $NP$ pass through valleys and over the Hindu-Kush mountain range, emphasising the importance of accurately representing the orography in this region in order to correctly reproduce the influence of the mid-latitude dynamics. Misrepresentation to the height and shape of the Hindu-Kush in numerical weather forecasting or climate models, as a result, for example, of too-coarse resolution, could significantly effect the pathways of mid-latitude air and result in unrealistic local moisture sources and sinks in the region.

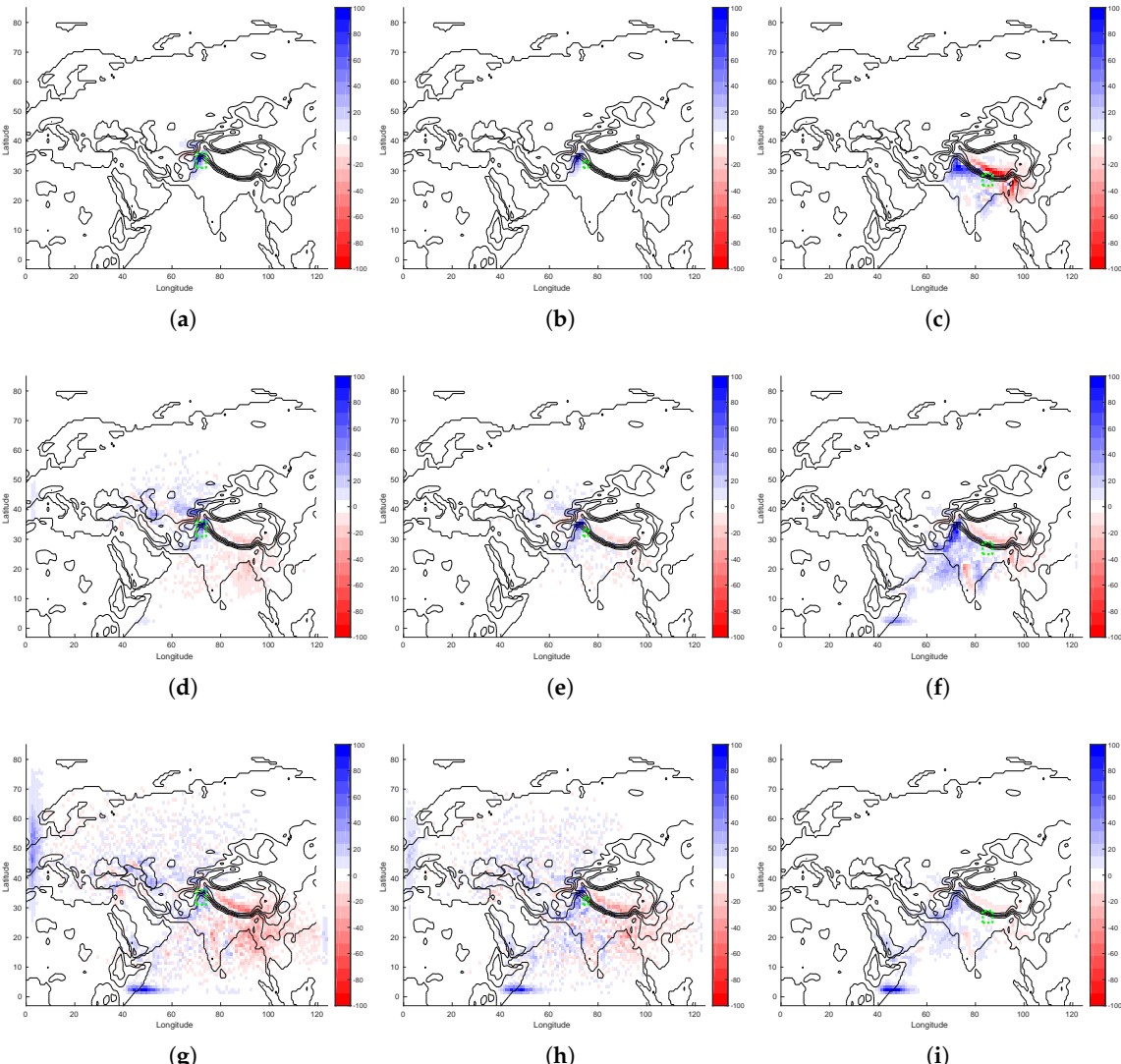

**Figure 4.** Summer (JJA) column integrated climatology (1980–2016), of the moisture sources and sinks, for all parcels over each 1° area, originating from: (**a**) *NP*, 2 days back in time, along the parcel trajectory; (**b**) *U*, 2 days back in time, along the parcel trajectory; (**c**) *K*, 2 days back in time, along the parcel trajectory; (**d**) *NP*, 6 days back in time, along the parcel trajectory; (**e**) *U*, 6 days back in time, along the parcel trajectory; (**f**) *K*, 6 days back in time, along the parcel trajectory; (**g**) *NP*, 10 days back in time, along the parcel trajectory; (**h**) *U*, 10 days back in time, along the parcel trajectory; (**i**) *K*, 10 days back in time, along the parcel trajectory. The area of interest is illustrated by a green dashed box. The colour bar scale shows the relative difference of the change in specific humidity.

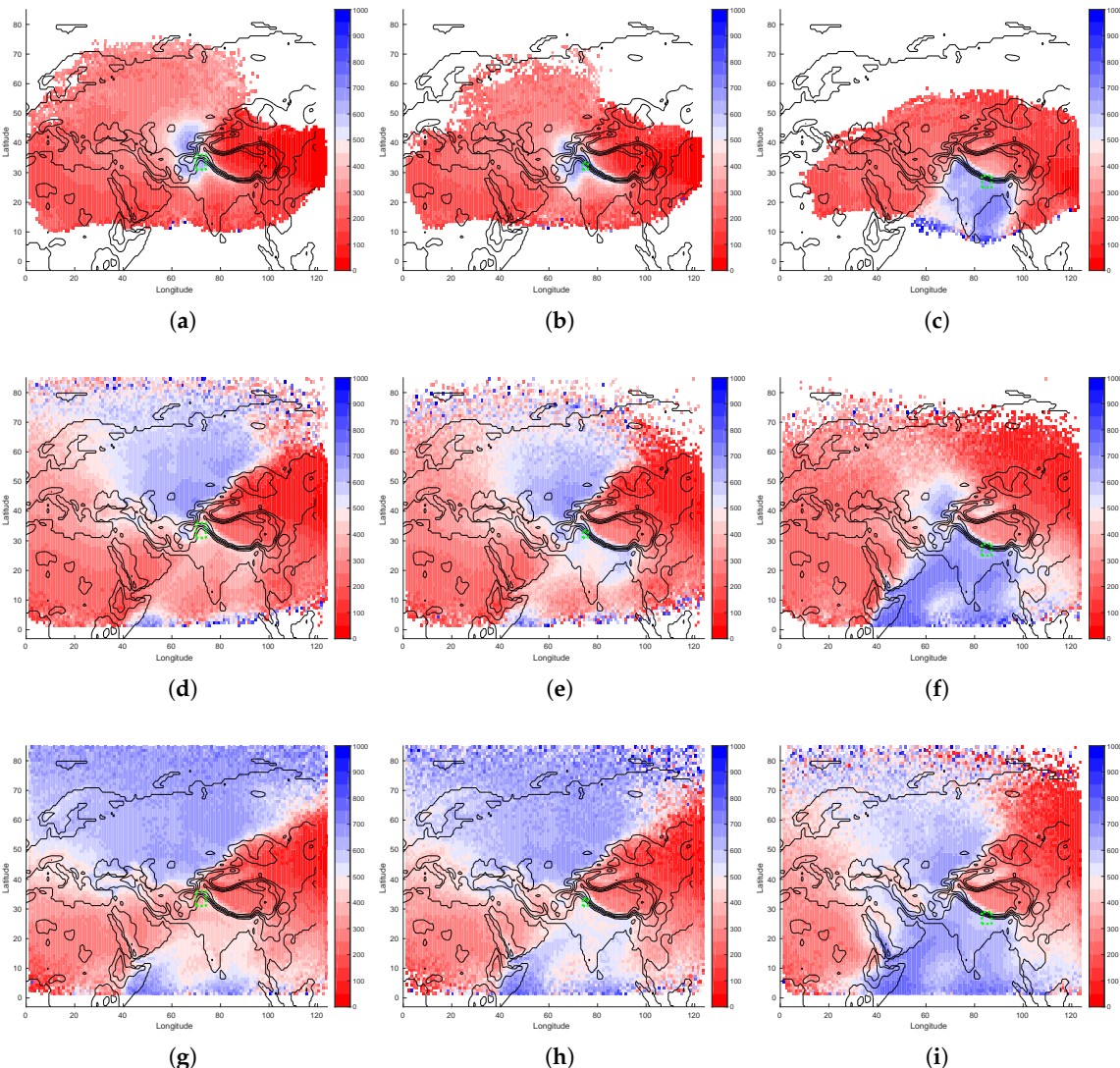

**Figure 5.** Summer (JJA) column mean climatology (1980–2016), of the parcel pressures, for all parcels over each 1° area, originating from: (**a**) *NP*, 2 days back in time, along the parcel trajectory; (**b**) *U*, 2 days back in time, along the parcel trajectory; (**c**) *K*, 2 days back in time, along the parcel trajectory; (**d**) *NP*, 6 days back in time, along the parcel trajectory; (**e**) *U*, 6 days back in time, along the parcel trajectory; (**f**) *K*, 6 days back in time, along the parcel trajectory; (**g**) *NP*, 10 days back in time, along the parcel trajectory; (**h**) *U*, 10 days back in time, along the parcel trajectory; (**i**) *K*, 10 days back in time, along the parcel trajectory. The area of interest is illustrated by a green dashed box.

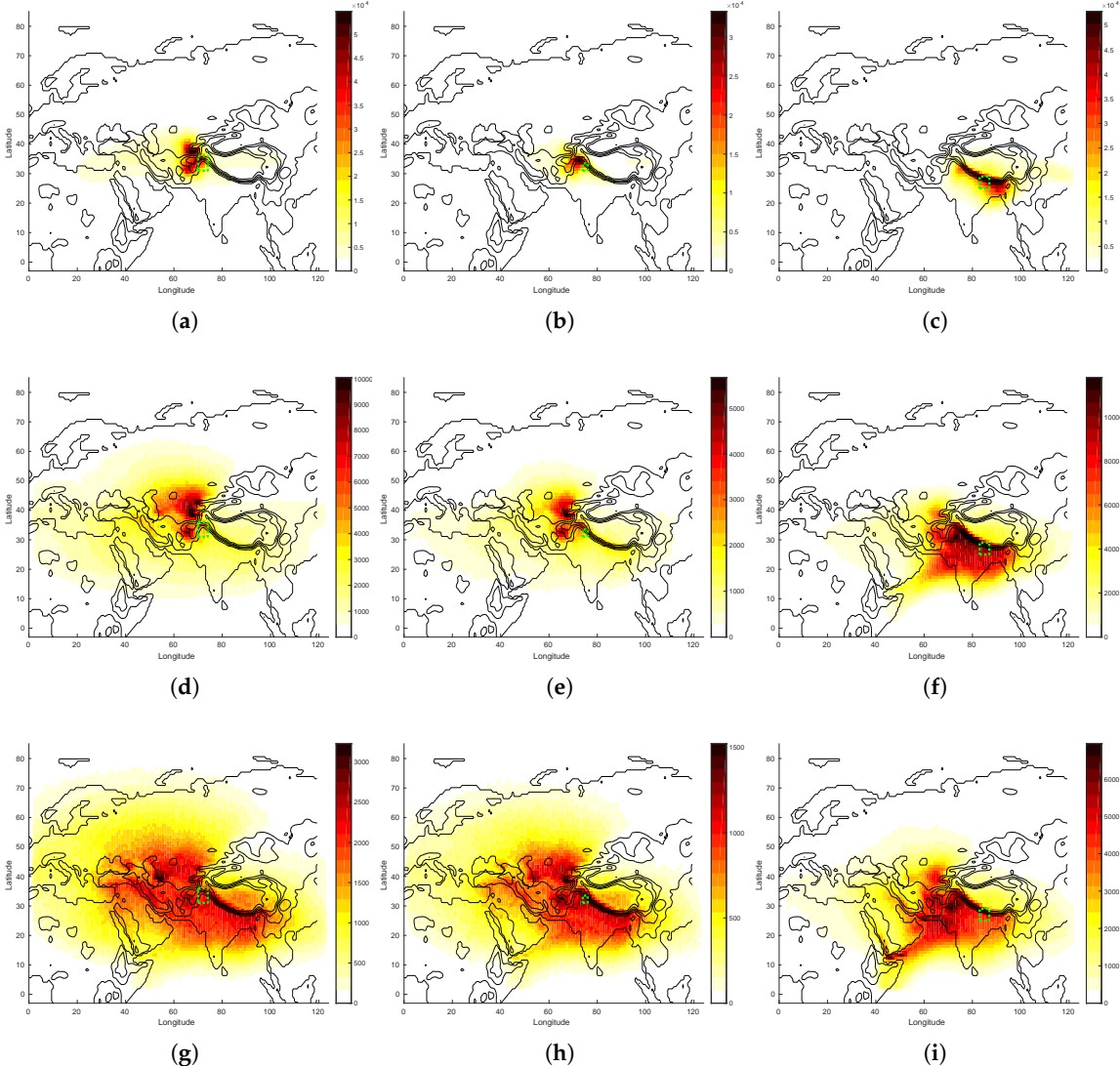

**Figure 6.** Summer (JJA) column integrated climatology (1980–2016), of the parcel density, for all parcels over each 1° area, originating from: (**a**) *NP*, 2 days back in time, along the parcel trajectory; (**b**) *U*, 2 days back in time, along the parcel trajectory; (**c**) *K*, 2 days back in time, along the parcel trajectory; (**d**) *NP*, 6 days back in time, along the parcel trajectory; (**e**) *U*, 6 days back in time, along the parcel trajectory; (**f**) *K*, 6 days back in time, along the parcel trajectory; (**g**) *NP*, 10 days back in time, along the parcel trajectory; (**h**) *U*, 10 days back in time, along the parcel trajectory; (**i**) *K*, 10 days back in time, along the parcel trajectory. The area of interest is illustrated by a green dashed box. If it is assumed that 36° is the dividing line between the tropics and extra-tropical regions and the number of parcels in each region is measured, we can obtain a rough measure of the relative influence of the extratropics on our regions of interest. From the 10-day back-trajectories, we measure that approximately 45%, 41%, and 20% of parcels originate from the extratropics for *NP*, *U*, and *K*, respectively.

### 3.1.2. Uttarakhand

The *U* region is situated just southwest of *NP* and, as a result, its climatology exhibits many of the same characteristics. We will, therefore, focus on the differences with respect to the *NP* region. The parcels en route to *U* exhibited a similar, though slightly reduced, mid-latitudinal influence. Once beyond the Hindu-Kush, parcels tended to gain moisture from the Indus basin (see Figure 4b,e,h), travelling in a south-eastward direction. It is rather notable that Figure 6b shows that, two days prior to reaching the target region, a vast majority of parcels were concentrated over *NP*.

In a departure from what we have found for the *NP* region, a stronger relevance of moisture sources from the Arabian sea, following slightly different pathways, was observed for the *U* region. The parcels en route to *U* tended to be travelling lower in the troposphere, according to Figure 5e,h, allowing for more moisture uptake from the boundary layer. The number of parcels originating from the Arabian sea was, however, not very large, in either relative or absolute terms (see Figure 6b,e,h). Parcels from the Bay of Bengal lost moisture on their way to *U*, while some evidence exists of moisture uptake in the Gangetic plain, indicating that recycling of water took place locally within the basin. In total, approximately 40% of parcels originated from the mid- and polar-latitude regions, 10 days back in time.

### 3.1.3. Kathmandu

For the *K* region, a much stronger sub-tropical influence over the mid-latitudes during the monsoon season was evident. Viewing the trajectories 6 and 10 days back, significant numbers of parcels were seen to extend only as far north as Northern Kazakhstan, while none of them originated from Russia, Western Europe, or the Mediterranean. In total, approximately 20% of parcels originated from the mid-latitudes or higher, 10 days back in time. Instead, parcels mostly from the Tropics, originating in the Red Sea, the Gulf of Aden, the Arabian Sea, and southeast India, reached *K* with high moisture content. The relevance of tropical circulation is clear. Nonetheless, Figures 4f,i and 6f,i indicate that the Indus Basin provided a key moisture source in this case, as well. This suggests that the strong influence of the monsoon weather systems over India were similarly important to the *K* region.

When looking two days back, there was also a clear divide from Northwest to Southeast, along the Southern Himalayan mountain range. Moisture gains for parcels moving to the *K* region were seen over Pakistan and areas south and west of the Southern Himalayan mountain range, along the Gangetic valley, while parcels over the Tibetan Plateau tended to precipitate. Note that no signature of the (scarce) Tibetan moisture was present in the two other locations, indicating the presence of a separate pathway of water transport for the *K* region. A large moisture sink also existed over Bangladesh and Meghalaya, where a large number of parcels precipitated en route to *K*, following the corresponding branch of the monsoon.

### 3.2. Moisture Sources and Sinks during the Winter

We now investigate the winter November–December–January (NDJ) seasonal climatology of moisture transport in the three locations described above, considering the whole 1980–2016 period. In this season, the monsoonal circulation was switched off and the mid-latitude westerlies were stronger than those during the summer. During winter, the majority of air parcels en route to *NP* and *U* transported moisture from the west. Figure 7a,b,d,e,g,h and the corresponding panels for Figures 8 and 9 show that moisture was picked up predominantly over the Mediterranean, Red, and Arabian Seas, 6–10 days prior to arrival. This is further clarified by the fact that the parcel densities for the 6 and 10 day back-trajectories, as shown in Figure 9d–i, indicate that most parcels passed over one of these regions en route to *NP*, *U*, and *K*. As opposed to summer conditions, the southward extent of the majority of moisture source events in winter occurred north of 10°; see the winter plots showing the parcel densities for parcels en route to *NP* in Figure 9a,b,d,e,g,h. The two-day back-trajectories show that parcels were concentrated over Iran and that its orography proved to be the largest moisture sink en route to *NP*.

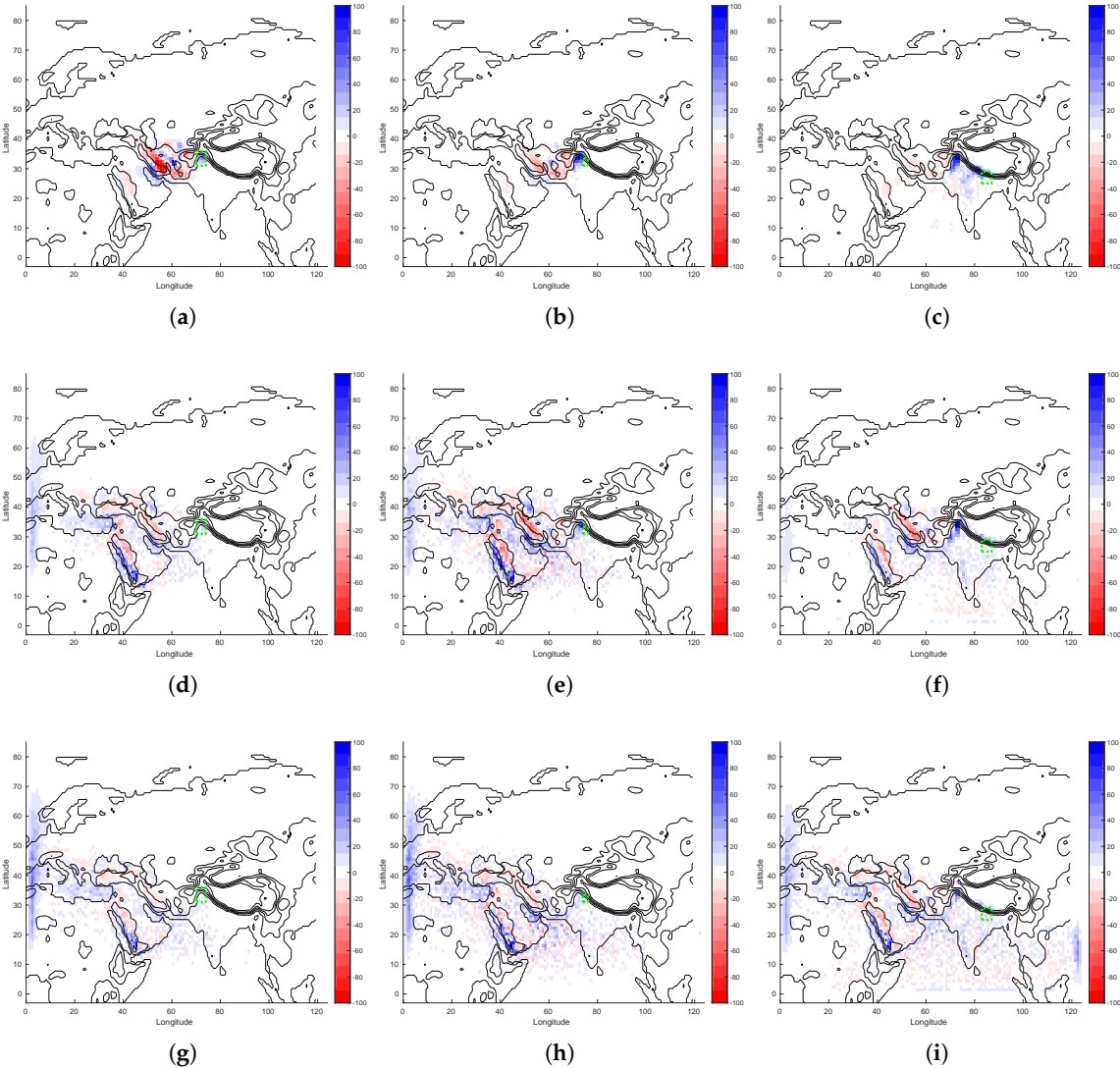

**Figure 7.** Winter (NDJ) column integrated climatology (1980–2016), of the moisture sources and sinks, for all parcels over each $1°$ area, originating from: (**a**) *NP*, 2 days back in time, along the parcel trajectory; (**b**) *U*, 2 days back in time, along the parcel trajectory; (**c**) *K*, 2 days back in time, along the parcel trajectory; (**d**) *NP*, 6 days back in time, along the parcel trajectory; (**e**) *U*, 6 days back in time, along the parcel trajectory; (**f**) *K*, 6 days back in time, along the parcel trajectory; (**g**) *NP*, 10 days back in time, along the parcel trajectory; (**h**) *U*, 10 days back in time, along the parcel trajectory; (**i**) *K*, 10 days back in time, along the parcel trajectory. The area of interest is illustrated by a green dashed box. The colour bar scale shows the relative difference of the change in specific humidity.

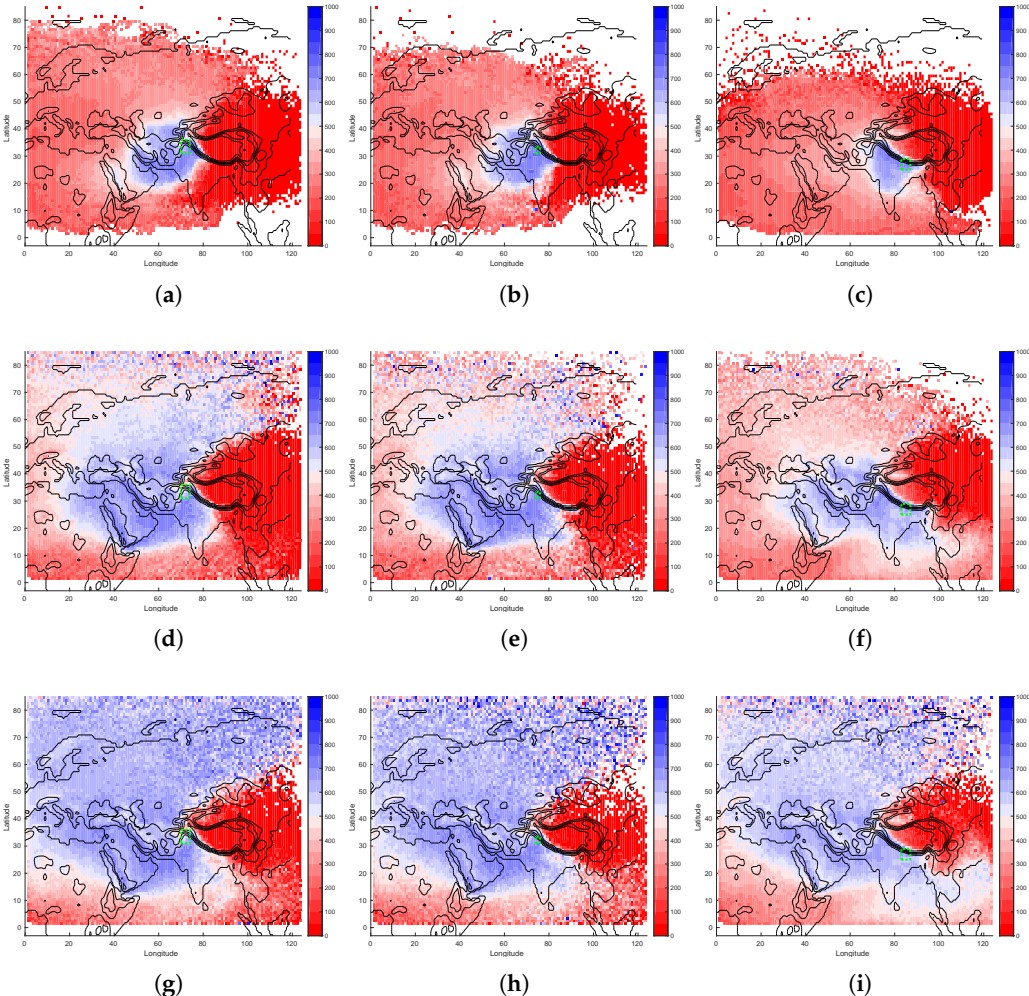

**Figure 8.** Winter (NDJ) column mean climatology (1980–2016), of the parcel pressures, for all parcels over each 1° area, originating from: (**a**) *NP*, 2 days back in time, along the parcel trajectory; (**b**) *U*, 2 days back in time, along the parcel trajectory; (**c**) *K*, 2 days back in time, along the parcel trajectory; (**d**) *NP*, 6 days back in time, along the parcel trajectory; (**e**) *U*, 6 days back in time, along the parcel trajectory; (**f**) *K*, 6 days back in time, along the parcel trajectory; (**g**) *NP*, 10 days back in time, along the parcel trajectory; (**h**) *U*, 10 days back in time, along the parcel trajectory; (**i**) *K*, 10 days back in time, along the parcel trajectory. The area of interest is illustrated by a green dashed box.

In general, all regions showed moisture uptake over the Red Sea and the Persian Gulf. Parcels en route to region *K* showed far less moisture uptake from the Mediterranean than to regions *NP* and *U*, but far more from the Bay of Bengal. Figure 7c,f,i show the winter sources and sinks en route to region *U*. Two days back in time, parcels en route to *NP* and *U* showed a strong moisture source band, running north to south across Pakistan. Directly northeast and southwest of this band, they also showed net sink regions in common. In contrast, *K* showed strong moisture uptake over west central India (see Figure 7). Parcels en route to region *K* also showed strong coastal uptake of moisture to the West of the Arabian sea. Across the 6 day step, parcels passing over Saudi Arabia tended to be a net source en route to the *K* region.

In comparison, the 6 and 10 day back-trajectories showed far fewer source regions close to the target regions in the winter months, compared to those of the summer. We note that this indicates that the air flow during the winter months was less stagnant, transporting parcels more quickly across South Asia due to the stronger zonal winds which tended to occur at this time of year. There was a fairly negligible amount of moisture coming from Russia (obviously due to the very cold and dry conditions there) during the winter. Instead, the trajectories of parcels tended to lie across a latitude range which covered Southern Europe through to Northern Africa. This, coupled with the fact (as shown in Figure 8) that the mean parcel height indicated a high proportion of parcels existing around the planetary boundary layer (PBL) height, naturally made the Mediterranean and Red Seas dominant moisture source regions. Note that evaporation in the Mediterranean sea peaked in winter and not in summer (as intuition would suggest), due to the strong and continued surface winds [53,54].

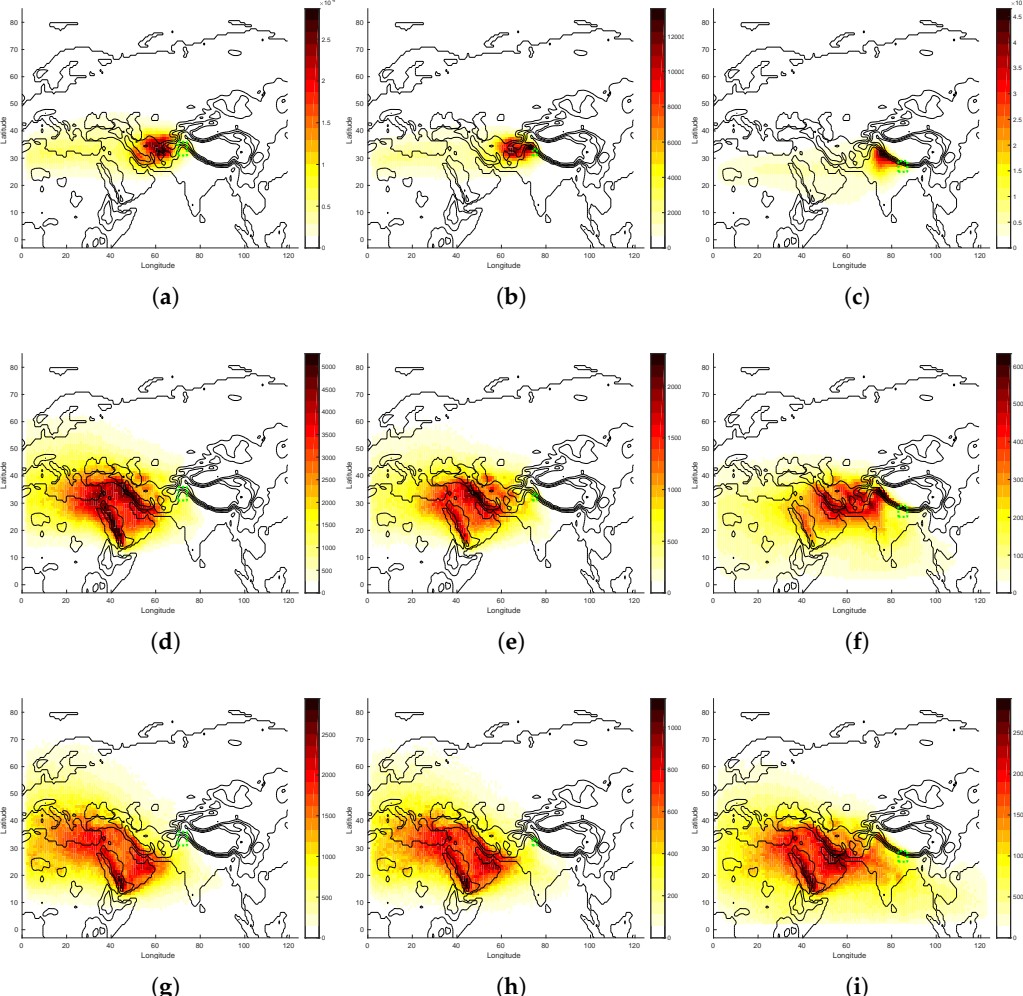

**Figure 9.** Winter (NDJ) column integrated climatology (1980–2016), of the parcel density, for all parcels over each 1° area, originating from: (**a**) $NP$, 2 days back in time, along the parcel trajectory; (**b**) $U$, 2 days back in time, along the parcel trajectory; (**c**) $K$, 2 days back in time, along the parcel trajectory; (**d**) $NP$, 6 days back in time, along the parcel trajectory; (**e**) $U$, 6 days back in time, along the parcel trajectory; (**f**) $K$, 6 days back in time, along the parcel trajectory; (**g**) $NP$, 10 days back in time, along the parcel trajectory; (**h**) $U$, 10 days back in time, along the parcel trajectory; (**i**) $K$, 10 days back in time, along the parcel trajectory. The area of interest is illustrated by a green dashed box.

These findings are in agreement with [55–57], who investigated the contribution of mid-latitudinal weather systems to precipitation in the Hindu-Kush region during winter. The first, [55], identified strong correlations between the North Atlantic oscillation (NAO) and winter precipitation in some specific locations. The latter two, [55–57], tracked western disturbances (WDs) from the mid-latitudes to the HKKH region and investigated the contributions made by such low-pressure systems to precipitation. The geographical region containing the tracks associated with these WDs matched well to the water pathways found here, confirming that many of our moisture source contributions from the Mediterranean, Persian Gulf, and Caspian sea were connected to the advection of such weather systems. The methodology deployed in this paper to track parcels did so indiscriminately and incorporated all weather systems en route to the *NP*, *U* and *K* regions. The differences between [56] and our study indicates that WDs were not strongly associated with the moisture sources of the southern half of the Red Sea and that an alternative mechanism of regional advection applied here.

## 4. Weather Extremes: Diagnosed Moisture Sources

Over longer durations of time, typical precipitation events occurring in such regions as Pakistan are crucial for maintaining water resources and glaciers against the threat of droughts; however, over intra-seasonal time scales, extreme weather event, leading to massive discharges of precipitation over a short period of time, have significant socio-economic impacts and the potential for loss of life. An important question to ask is whether these same areas which contribute to the average climatological rainfall also correspond to the sources of extreme events, or whether there is something dynamically peculiar about the extreme events we want to study. Here, we select the most significant rainfall events to have befallen the *NP*, *U*, and *K* regions, and consider the moisture paths and sources leading up to those events.

### 4.1. Northern Pakistan Floods of July 2010

Figures 10 and 11 show all parcels en route to *NP* for the periods 22 July 2010 and 28 July 2010, respectively. In the case of Figure 12 (the event in *U* on 16 June 2013) and Figure 13 (the event in *K* on 13 August 2017), the left and right columns show each parcel, coloured by their pressure and specific humidity. The top, middle, and bottom rows give the parcel positions 2, 6, and 10 days back in time, respectively.

In this way, viewing the parcels in the left columns of Figures 10 and 11 illustrates the distinct separation between the paths travelled by parcels which were high and low in the troposphere. In Figure 11b, we can see only those parcels which contributed to the precipitation event on 28 July 2010. We see that these dry parcels in the upper troposphere (denoted in red) over the Tibetan Plateau did not, in fact, contribute directly to the precipitation event.

Conversely, the parcels lower in the troposphere (denoted in blue), came from across central and northern India, Bangladesh, Pakistan, the Arabian sea, Russia, and northern Europe. Due to their proximity to the ground, these parcels took up moisture mostly from the boundary layer and transported it, remaining mostly below the 500hPa height for the majority of these parcels. The moisture uptakes within the boundary layer can be linked directly to moisture evaporating from the surface below, as the boundary layer is considered to be well-mixed. Compared to the climatology of parcels travelling to *NP*, the parcels arriving from the Indian subcontinent were travelling lower in the troposphere, providing more favourable conditions to gain moisture from the boundary layer.

Figures 10b,d,f and 11b,d,f show the origin of the moisture contributions for parcels associated with the precipitation events on 22 and 28 July, respectively. From the north, parcels traveled east over northern Russia, moving/meandering around the blocking high, and travelling south towards the $NP$ region. These parcels were relatively dry and few contributed directly to the precipitation events on the 22nd and 28th.

We chose to show both of the dates for the July flooding events, in order to illustrate how the sources of moisture over this period varied. For 22 July, a majority of the parcels that contributed to the precipitation events originated from and gained moisture over the Arabian sea and northern India. The parcels traveled from the Arabian sea in a north-eastward direction, passing over the Northwest coast of India before turning Northwest and travelling toward $NP$ along the Himalayan foothills. These findings are consistent with those of [23], where it was showed that moisture transport acts similarly en route to $NP$. For the 28th, the moisture parcels travelling to $NP$ originated from the Arabian Sea, central and southern India, and the the Bay of Bengal; however, the parcels travelling from the Arabian Sea took a much wider path, reaching $NP$ by travelling through central India and the Bay of Bengal.

The parcels travelling from Russia were, for the most part, relatively dry and only picked up recycled moisture locally within the Hindu-Kush region. This moisture was supplied by the parcels travelling over India and the Arabian sea under monsoon conditions to $NP$, but which precipitated before reaching $NP$. For both 22 and 28 July, it is clear that there was a lower-level convergence between parcels travelling from Russia associated with the blocking high and those travelling over the Arabian Sea and the Gangetic plain. These results support previous studies, such as [21], most of which applied Eulerian techniques to study this problem and which strongly implied the importance of the role played by the tropical–extratropical interaction; being necessary in triggering the 2010 Pakistan extreme rainfall event. More specifically, the results here support the implications of [21] discussed previously, that the low-level convergence of the deep trough bringing dry air mass from Russia and warm moist air from the Arabian sea and the Indian subcontinent. The strength of the result presented here is that the trajectories of the air parcels were directly shown to converge to the Pakistan flood region and coincided with the blocking high and trough which extended down to $NP$. Furthermore, the associated warm moisture transports seen here were also evident in [23,58].

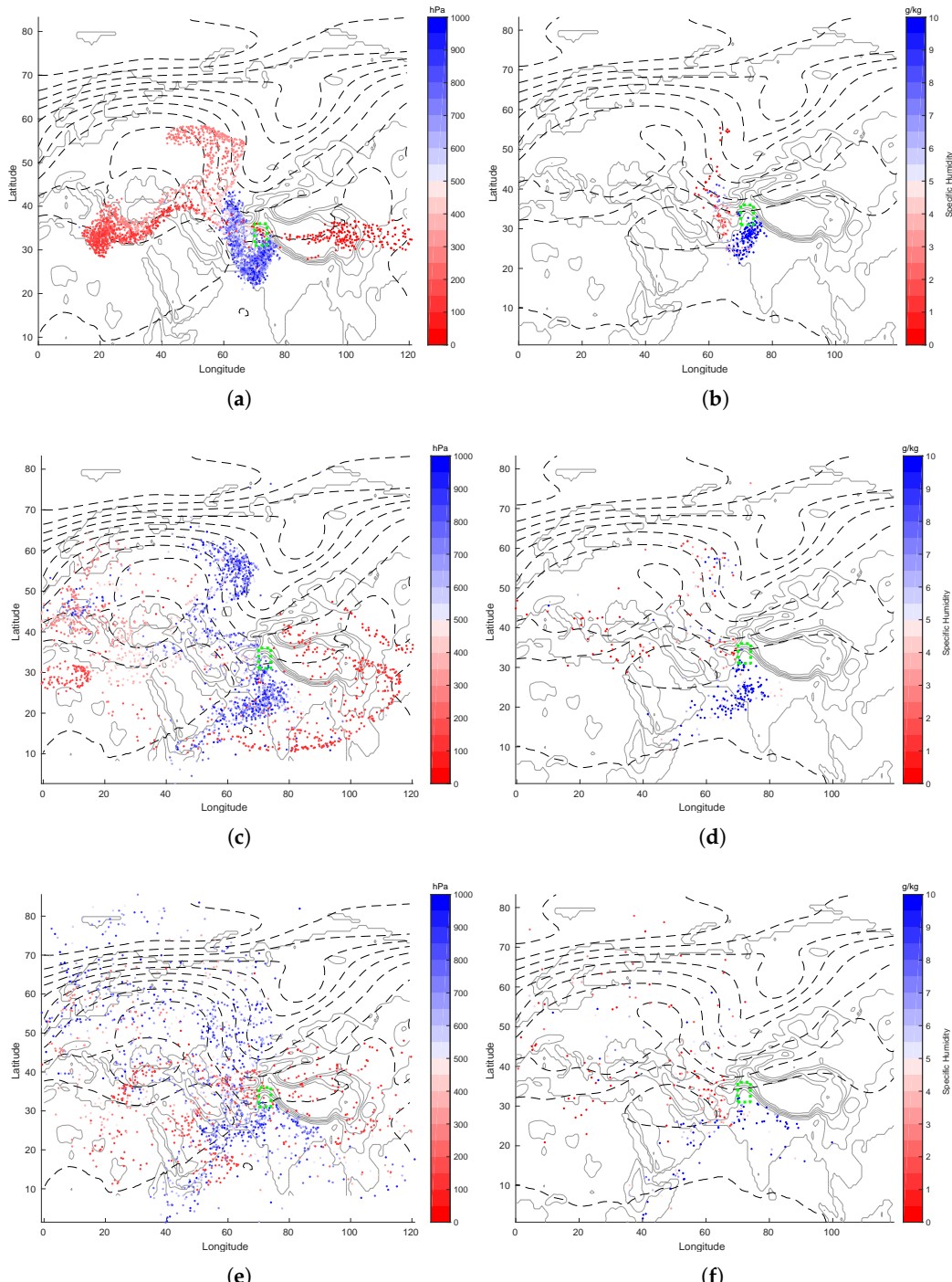

**Figure 10.** All back-trajectories emanating from *NP* on 22 July 2010. (**a**) parcel pressure, 2 days back in time, along the parcel trajectory; (**b**) parcel pressure, 2 days back in time, along the parcel trajectory; (**c**) parcel pressure, 6 days back in time, along the parcel trajectory; (**d**) specific humidity, 6 days back in time, along the parcel trajectory; (**e**) specific humidity, 10 days back in time, along the parcel trajectory; (**f**) specific humidity, 10 days back in time, along the parcel trajectory. The dashed lines indicate the 5 and 2 day means of 500 mb geopotential height for the left and right columns, respectively.

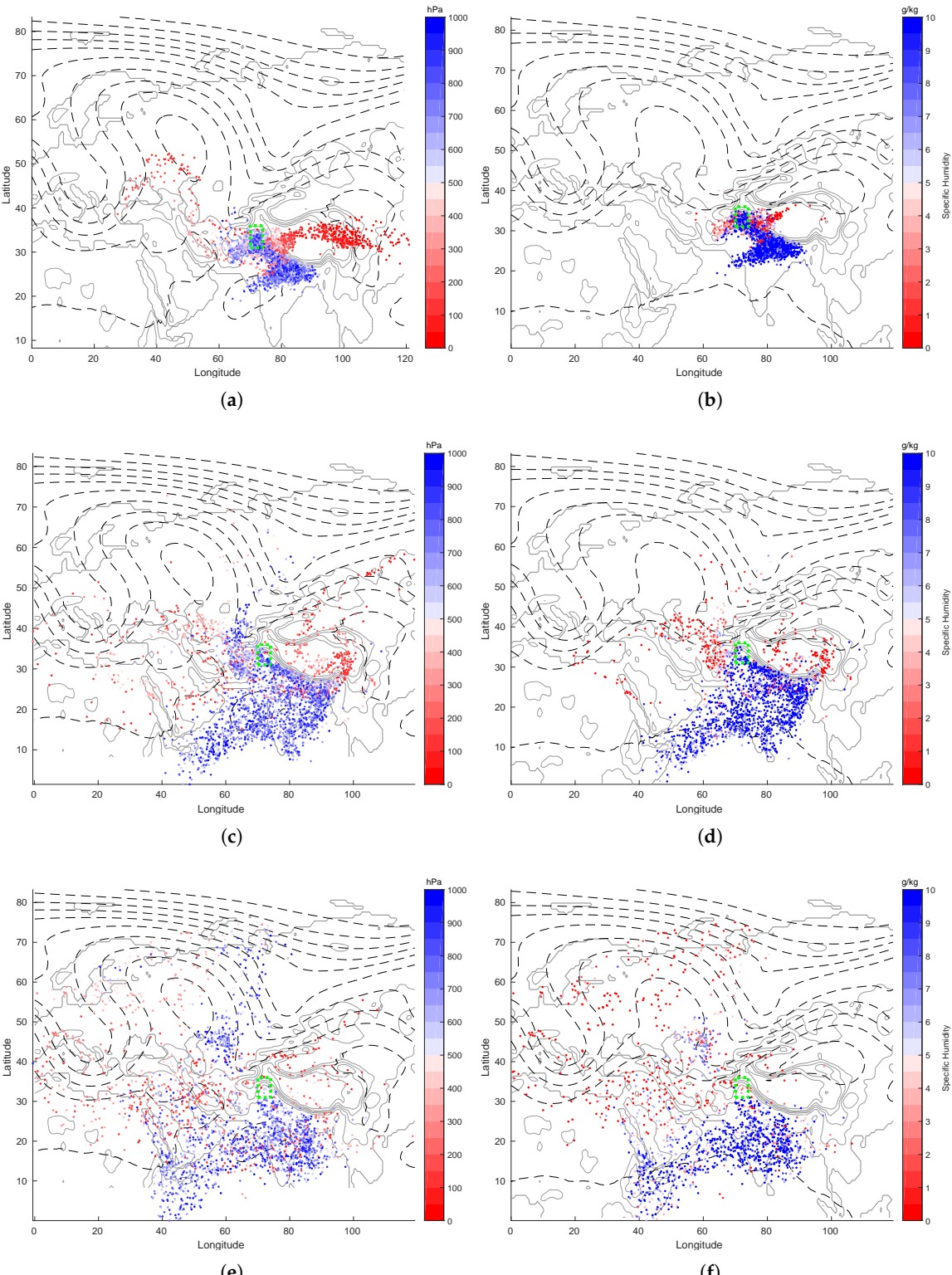

**Figure 11.** All back-trajectories emanating from *NP* on 28 July 2010. (**a**) parcel pressure, 2 days back in time, along the parcel trajectory; (**b**) parcel pressure, 2 days back in time, along the parcel trajectory; (**c**) parcel pressure, 6 days back in time, along the parcel trajectory; (**d**) specific humidity, 6 days back in time, along the parcel trajectory; (**e**) specific humidity, 10 days back in time, along the parcel trajectory; (**f**) specific humidity, 10 days back in time, along the parcel trajectory. The dashed lines indicate the 5 and 2 day means of 500 mb geopotential height for the left and right columns, respectively.

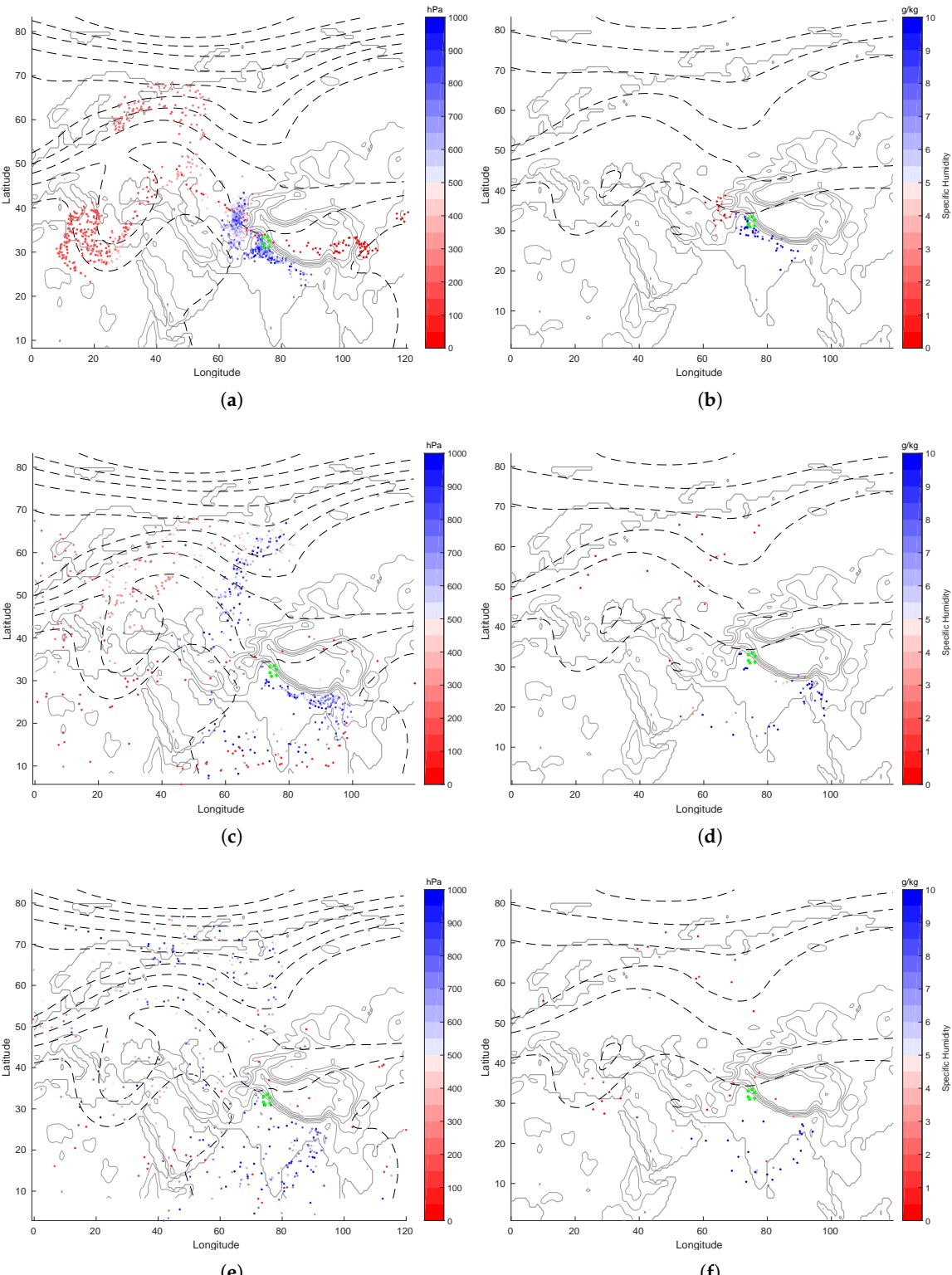

**Figure 12.** All back-trajectories emanating from *U* on 16 June 2013. (**a**) parcel pressure, 2 days back in time, along the parcel trajectory; (**b**) parcel pressure, 2 days back in time, along the parcel trajectory; (**c**) parcel pressure, 6 days back in time, along the parcel trajectory; (**d**) specific humidity, 6 days back in time, along the parcel trajectory; (**e**) specific humidity, 10 days back in time, along the parcel trajectory; (**f**) specific humidity, 10 days back in time, along the parcel trajectory. The dashed lines indicate the 5 and 2 day means of 500 mb geopotential height for the left and right columns, respectively.

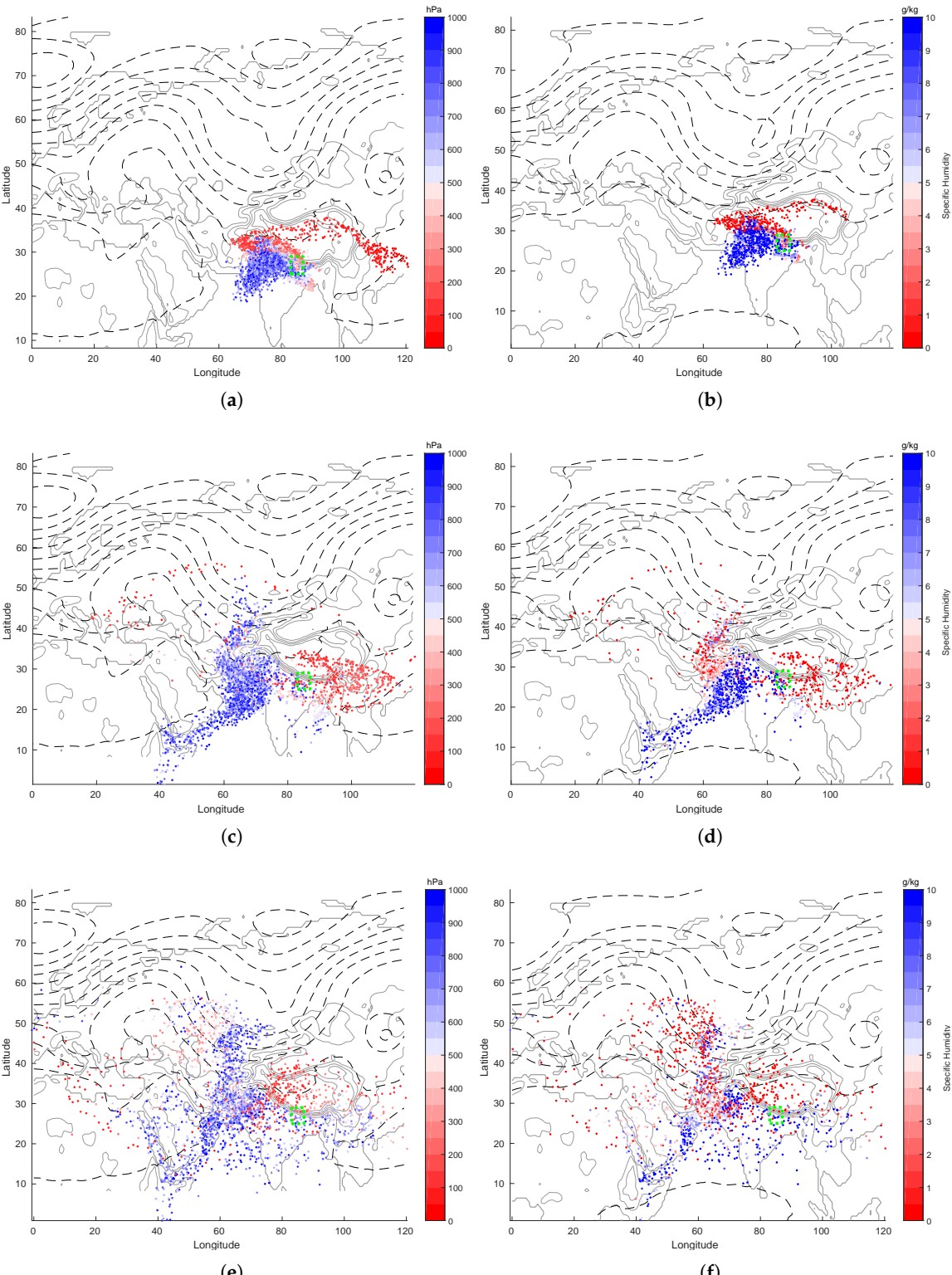

**Figure 13.** All back-trajectories emanating from *K* on 13 August 2017. (**a**) parcel pressure, 2 days back in time, along the parcel trajectory; (**b**) parcel pressure, 2 days back in time, along the parcel trajectory; (**c**) parcel pressure, 6 days back in time, along the parcel trajectory; (**d**) specific humidity, 6 days back in time, along the parcel trajectory; (**e**) specific humidity, 10 days back in time, along the parcel trajectory; (**f**) specific humidity, 10 days back in time, along the parcel trajectory. The dashed lines indicate the 5 and 2 day means of 500 mb geopotential height for the left and right columns, respectively.

### 4.2. Uttarakhand Floods of June 2013

Figure 12 shows all parcels en route to *U* for the period leading up to 16 June 2013. Another recent, and rather significant, precipitation event leading to major flooding occurred during June 2013 in *U*. Looking at the parcels back-tracked by 10 days, from the north there was no clear concentration of parcels to mark the main origin or path for the parcels to take. Nevertheless, many parcels were drawn in from Western Europe and Russia. From the south, parcels traveled over the Arabian Sea, moving east-northeast over southern India and the Bay of Bengal. Six days back, the parcels passed over the Bangladesh wetlands before turning in an northwest direction. We note this was a departure from the climatology, which showed relatively few numbers of parcels passing over Bangladesh en route to *U*. Two days back, the moist monsoonal parcels travelling northwest over the Gangetic Valley were mostly in the vicinity of *U*. Meanwhile, low-humidity parcels from the upper troposphere converged from the north-west. This is especially evident in the half day back-trajectory positions.

The above findings agree with those of [21,23,58,59], showing the key characteristic which led to these extreme events was a result of atypical coupling between mid-latitudinal and monsoonal weather systems, which was realised through the convergence of fast-moving, low-level air from the northwest and low-level moist air from the south-east against the mountainous orography, which resulted in strong updrafts, driving the observed extreme precipitation events. It is evident, here, that the moist parcels which travelled over the Arabian Sea and the Indian subcontinent to *NP* and *U* were anomalously low in the troposphere during these extreme events, meaning these parcels were in a state receptive to moisture accumulation and resistant to precipitation. The reverse was true for the climatologies shown in Section 3, where air parcels in the sub-tropic region, en route to *NP* and *U*, were typically in the mid-troposphere. Therefore, there was a key moisture supply that only existed during the extreme events, which was likely a direct consequence of the low-level convergence associated with the trough of the mid-latitudinal weather system. Conversely, while the extreme events studied here indicated that the low-level travel of moist parcels was crucial and that the ultimate source of the moisture was similar, the latitudinal–longitudial path taken is likely a less determinative factor, as the moisture availabilities over the Arabian sea and Gangetic plain are both normally high during the monsoon season.

### 4.3. Kathmandu Floods of August 2017

Figure 13 shows all parcels en route to *K* for the period leading up to 13 August 2017. Moisture parcels predominately travelled from the Arabian sea, at low levels. Some relatively dry air parcels were observed to travel from Russia, through Kazakhstan and then the Indus basin, where they took up moisture en route to *K*. In contrast to the *NP* and *U* regions, both the climatological and extreme perspectives appeared to be consistent with one another, and the event is therefore representative of the kind of heavy precipitation event typically occurring during active periods of the monsoon. This further demonstrates how the influence of airflow from the mid-latitudes on the climatologies of *NP* and *U* implies that these regions have a dryer annual mean climate and that extreme precipitation periods represent special and atypical cases of the monsoon extending more than usual, albeit as a result of mid-latitudinal waves.

### 4.4. Summer Extremes: Sources and Sinks

We have considered, above, three specific case studies of weather systems which led to extreme events and how those cases differed from the climatology. While we cannot show, here, an exhaustive list of examples of extreme events on an individual basis, it is instructive to analyse and show the pathways of a subset of the climatology, based on a number of the strongest precipitation events. Here, we analyse the climatology of precipitation extremes by using the APHRODITE dataset to select the 20 days with the most significant *JJA* rainfall over the *NP*, *U*, and *K* regions between 1980–2007. Figures 14–16 show the moisture sources and sinks, mean parcel height, and parcel density, respectively.

For *NP* and *U*, local moisture pickup over the Indus Basin was again found to be the largest contributing factor (Figure 14a,b), while moisture sources originating across the Arabian Sea, India, Bangladesh, the Bay of Bengal, and the Gangetic plain also played a significant role (Figure 14d,e,g,h). This is supported by the fact that the mean height of parcels en route to these regions was low in the troposphere (Figure 15a,b,d,e,g,h), a key distinction from the climatology already reported in the case studies for the *NP* and *U* regions. Furthermore, a greater proportion of these parcels originated from the subcontinent, as opposed to the mid-latitudes (Figure 5d,e,g,h). This was especially notable for case over northeast India and Bangladesh, 6 days back in time.

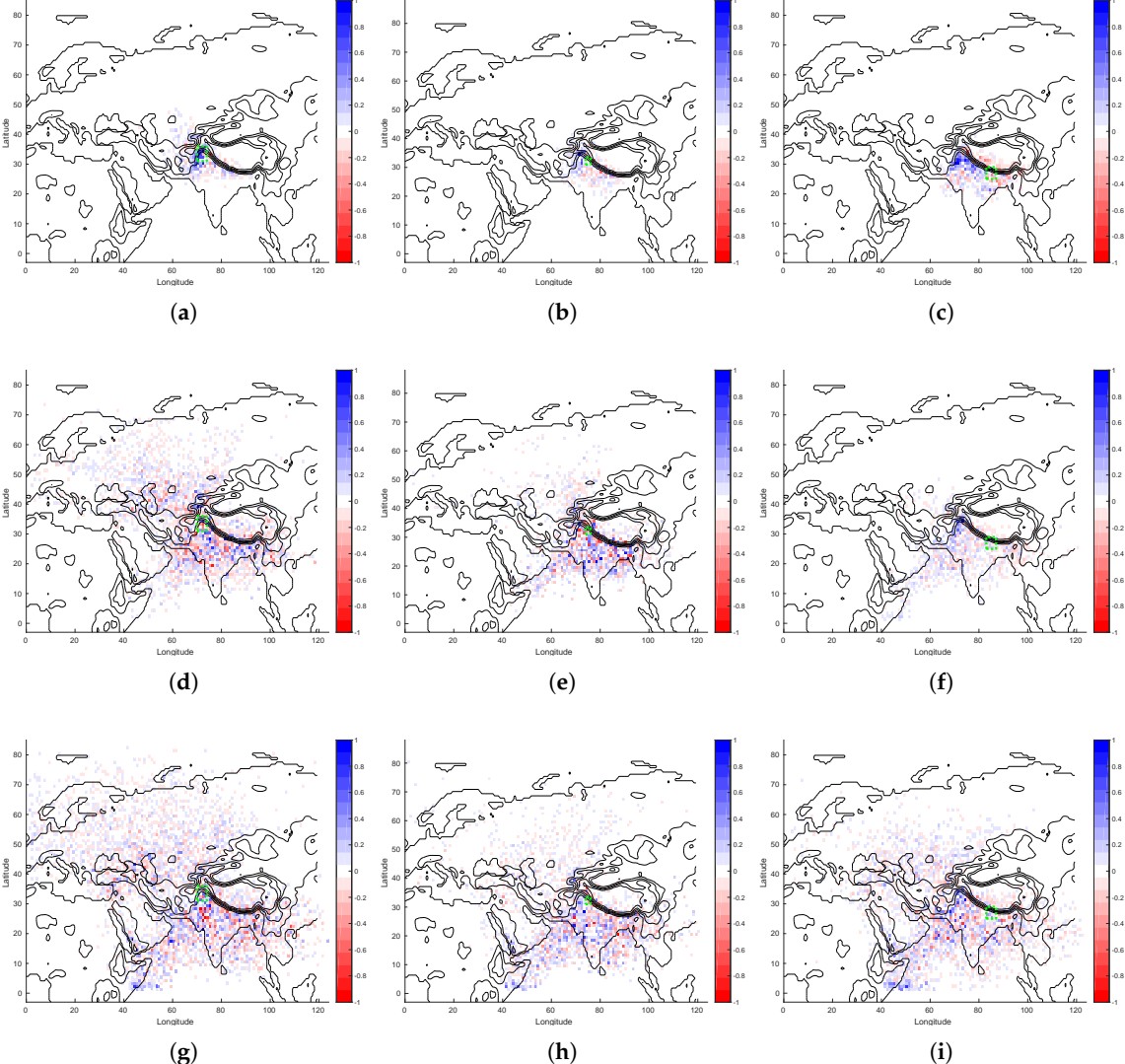

**Figure 14.** For the 20 strongest precipitation events for each region of interest, the column integrated moisture sources and sinks, for all parcels over each $1°$ area, originating from: (**a**) *NP*, 2 days back in time, along the parcel trajectory; (**b**) *U*, 2 days back in time, along the parcel trajectory; (**c**) *K*, 2 days back in time, along the parcel trajectory; (**d**) *NP*, 6 days back in time, along the parcel trajectory; (**e**) *U*, 6 days back in time, along the parcel trajectory; (**f**) *K*, 6 days back in time, along the parcel trajectory; (**g**) *NP*, 10 days back in time, along the parcel trajectory; (**h**) *U*, 10 days back in time, along the parcel trajectory; (**i**) *K*, 10 days back in time, along the parcel trajectory. The area of interest is illustrated by a green dashed box. The colour bar scale shows the relative difference of the change in specific humidity.

The generalised extremes of *NP* and *U* shared the same characteristics as the case studies discussed in Sections 4.1 and 4.2, thus showing key differences from the features of the summer

climatology. Conversely, the *K* region again confirmed the suggestion of the August 2017 flood event which hit *K* (as discussed in Section 4.3), that the moisture pathways of the summer climatology were consistent with those for the precipitation extremes. Thus, we observed moisture sources more often originating in the subcontinent, rather than from the mid-latitude regions, with moisture sources mostly originating over the Indus Basin and the Gangetic plain (north of *K*) for $(E - P)_2$, but also from the Arabian Sea, Persian Gulf, and Bay of Bengal. Some discernable differences were seen, however, with almost no signal coming from the Red Sea. Furthermore, the number of parcels and moisture sources originating from the mid-latitudes were slightly greater 6 and 10 days back. This is also likely related to the fact, that 2 days back, we observed a greater parcel density along the Gangetic Plain north of *K*, but less so south of *K*; which is inverse to what is observed in the climatology in Figure 6c.

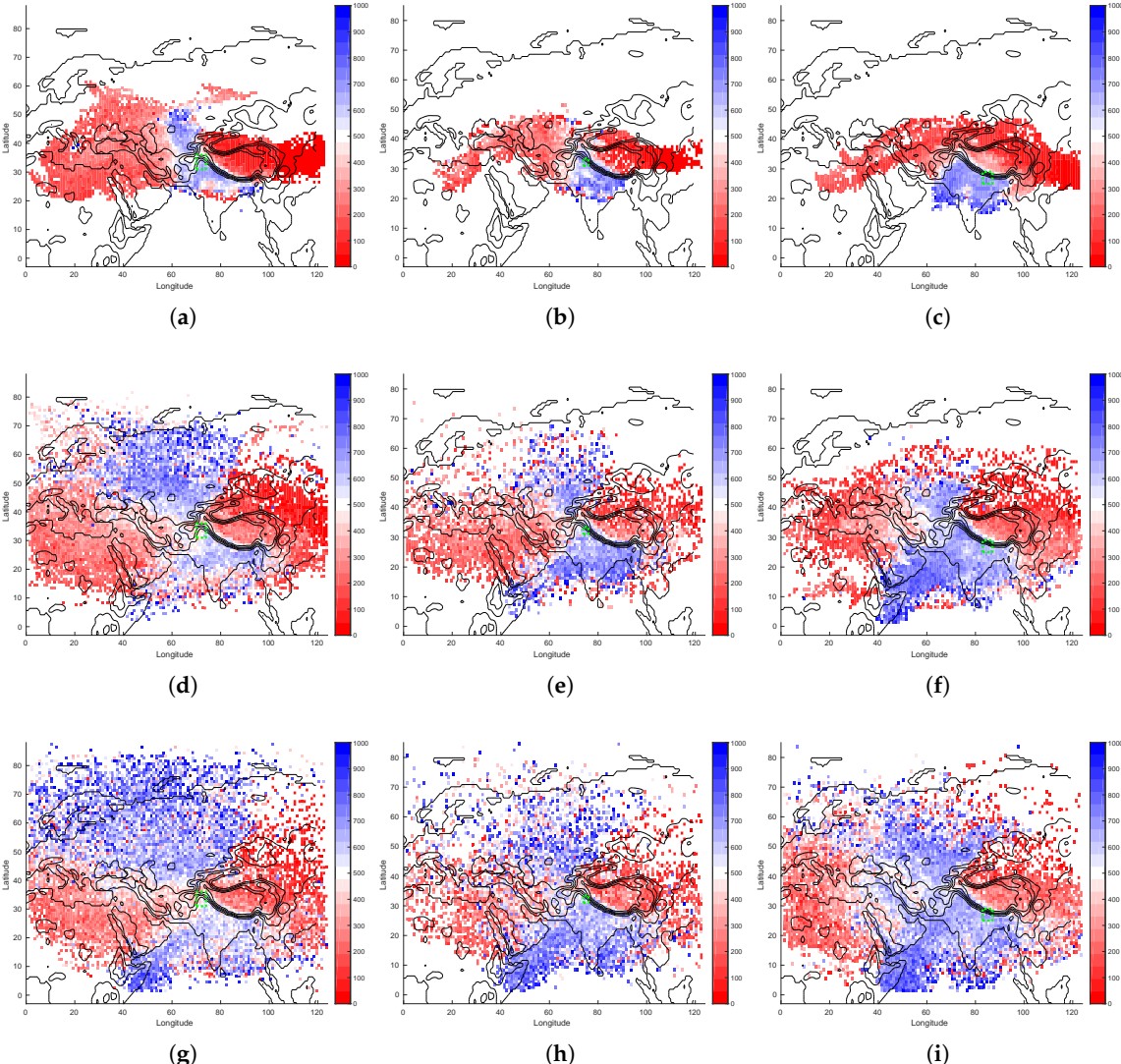

**Figure 15.** For the 20 strongest precipitation events for each region of interest, the parcel pressures, for all parcels over each $1°$ area, originating from: (**a**) *NP*, 2 days back in time, along the parcel trajectory; (**b**) *U*, 2 days back in time, along the parcel trajectory; (**c**) *K*, 2 days back in time, along the parcel trajectory; (**d**) *NP*, 6 days back in time, along the parcel trajectory; (**e**) *U*, 6 days back in time, along the parcel trajectory; (**f**) *K*, 6 days back in time, along the parcel trajectory; (**g**) *NP*, 10 days back in time, along the parcel trajectory; (**h**) *U*, 10 days back in time, along the parcel trajectory; (**i**) *K*, 10 days back in time, along the parcel trajectory. The area of interest is illustrated by a green dashed box.

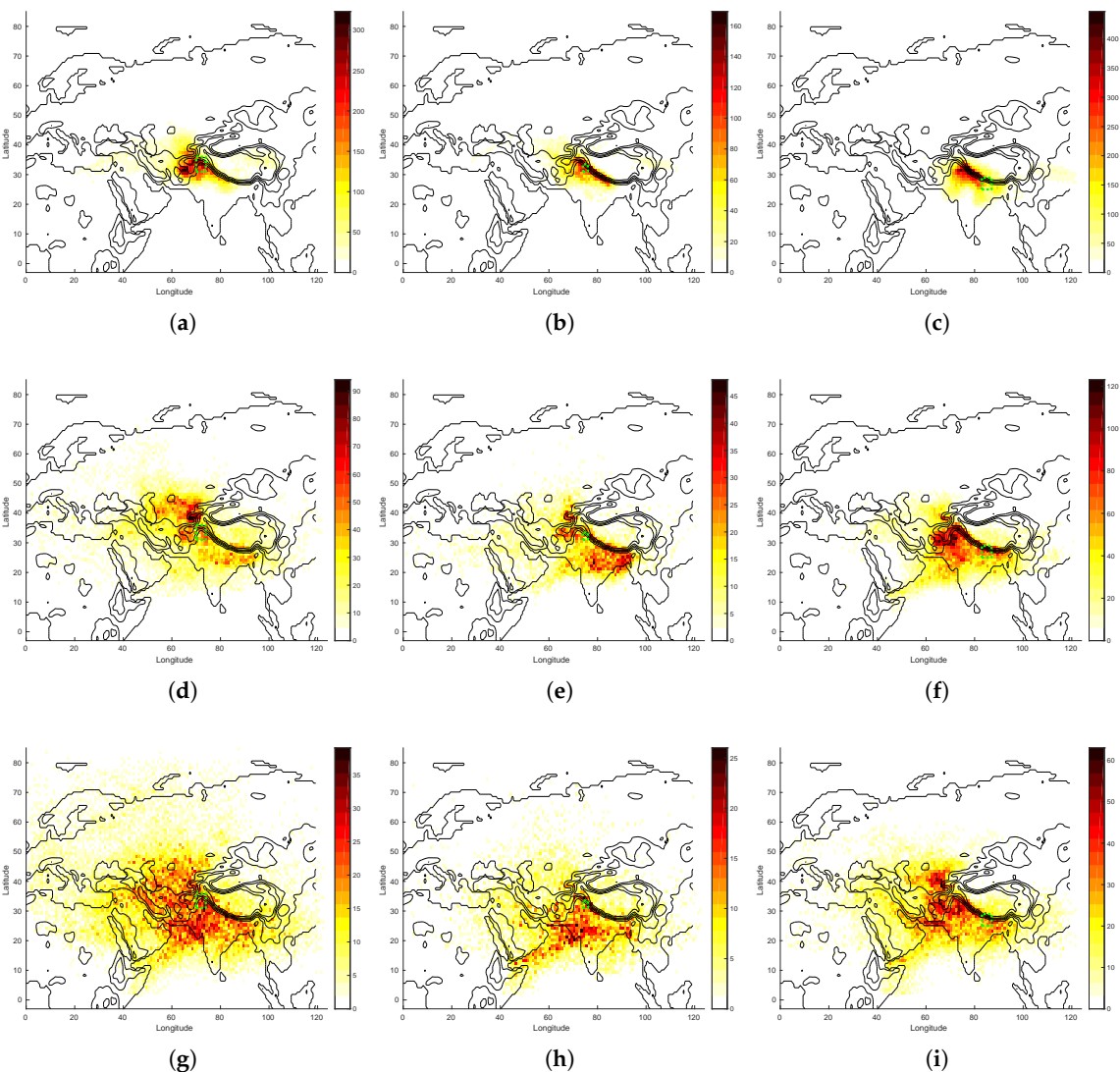

**Figure 16.** For the 20 strongest precipitation events for each region of interest, the parcel density, for all parcels over each 1° area, originating from: (**a**) *NP*, 2 days back in time, along the parcel trajectory; (**b**) *U*, 2 days back in time, along the parcel trajectory; (**c**) *K*, 2 days back in time, along the parcel trajectory; (**d**) *NP*, 6 days back in time, along the parcel trajectory; (**e**) *U*, 6 days back in time, along the parcel trajectory; (**f**) *K*, 6 days back in time, along the parcel trajectory; (**g**) *NP*, 10 days back in time, along the parcel trajectory; (**h**) *U*, 10 days back in time, along the parcel trajectory; (**i**) *K*, 10 days back in time, along the parcel trajectory. The area of interest is illustrated by a green dashed box.

## 5. Conclusions

In this study, we have investigated the moisture pathways to Northern Pakistan (*NP*), Uttarakhand (*U*), and Kathmandu (*K*) on climatological time scales, from 1980–2016, during the monsoon and winter seasons, as well as making a comparison of the general behaviour to that of the specific case studies of extreme precipitation events in these regions during July 2010, June 2013, and August 2017, respectively. Furthermore, we examined a subset of the climatology which provided generalised pathways for the 20 most intense daily extreme events to occur over each region during the summer months.

In doing so, we have also shown the relative influence between extra-tropical and tropical weather systems for the climatology, as we move east along the Gangetic plain at the foot hills of the HKKH. This study was performed from a Lagrangian perspective, which provided a complementary approach to

the more-commonly used Eulerian methods. Using a modified version of FLEXPART, the atmosphere was divided up into even mass parcels, which were subsequently tracked along their trajectories backwards in time (forced by the ECMWF wind fields), while recording changes to parcel properties such as pressure, specific humidity, and temperature. This, then, meant that we could identify changes to the water content of each parcel at each time step, allowing us to calculate the regional moisture sources and sinks over the whole climatological period in question.

From a climatological perspective, during the monsoon, it was found that the largest source of moisture to $NP$ was from local regions within the previous two days of a parcel's journey. A significant amount of the moisture gain is likely due to recycling from the Indus basin, as a result of moisture deposited over the Hindu-Kush, where the mountains act as a significant barrier to moisture en route to $NP$. Additional moisture sources come from locations, such as the Arabian Sea off the coast of Africa, 10 days back in time. A notable moisture source was also observed to come from Eastern Europe and Russia, a region where many parcels travel from, en route to $NP$. A key pathway was over and through the valleys of the Hindu-Kush, suggesting the importance of accurately resolving the orography in achieving an accurate representation of precipitation patterns over $NP$ and the Northwestern Indian regions. To the west, parcels coming through India and Nepal tended to precipitate before reaching Pakistan, in agreement with the fact that it tends to be far dryer than the rest of the South Asian monsoon region. For $U$, the characteristic pattern of moisture sources was similar to $NP$, with the largest moisture sources being over the Indus Basin two days back, although the influence of sub-tropical weather systems, relative to those of the mid-latitudes, was marginally greater. As expected, moving further east to $K$, this was even more the case, where the influence of sub-tropical weather systems dominated over those of of the mid-latitudes. The main sources of moisture here were found to be from the Indus Basin, the Gangetic Valley, Arabian Sea, Indus Basin, and south east India. Further afield, some moisture was gained from the Red Sea and the coastal region of Somalia, attributable to the Somalian jet. Overall, we were able to clearly illustrate the decreasing (increasing) influence of the mid-latitudinal (sub-tropic) weather systems on the climatology as we move East across the Gangetic Plain. A schematic representation of the main water pathways for the winter season is given in Figure 17.

During the winter months, the main moisture sources of $NP$ and $U$ can be attributed to the Red, Mediterranean, and Caspian Seas, the Persian Gulf, and the Gulf of Aden, some of which are associated with (WDs), as discussed in previous studies ([56,57]), al though the analysis presented here can not be broken down into the relative contributions of the types of weather systems advecting the moisture. Along the two day back-trajectory, we observed an especially strong sink of moisture over the mountainous regions of Iran, indicting that the potential source of moisture for recent flooding events in this region may be related to moisture originating from the Mediterranean Sea. As for the $K$ region, the patterns of the moisture sources were somewhat more local, resulting from a weaker influence of the zonal circulation of the mid-latitudes.

We also considered three major precipitation events, which led to catastrophic flooding. The first was in July 2010 and the second was in June 2013. The precipitation event of 2010 occurred under the backdrop of a persistent blocking over Russia, which lasted about two months; while the 2013 flood occurred under a trough, which formed and dissipated over a period of about five days. Though the development of these weather systems were rather different, both left a trough in proximity to the Hindu-Hush Himalayan region, providing conditions for low-level convergence between the air mass travelling from the North (over Russia) and that from the seas around the Indian subcontinent. What was consistent, between all dates for both events considered here, was, within two days of the precipitation event, a key narrow moisture pathway existed, where parcels traveled north-west en route to $NP$.

Another key aspect to the 2010 and 2013 flooding events was the fact that much of the moisture from the Indian subcontinent, which would normally not converge on to $NP$, did so, in combination with a significant supply of air flow from Russia, which drove the vertical motion of the humid

air mass in the presence of the orographic features defining the region. In the typical climatology, the air parcels travelling from the Indian subcontinent usually travel high in the troposphere, and the majority of parcels will not gain moisture from the boundary layer and will, instead, tend to precipitate. However, this was not the case during the extreme events analysed here, which indicated that the moist parcels travelled over the Arabian Sea and the India subcontinent, to *NP* and *U*, anomalously low in the troposphere, meaning that those parcels were in a state receptive to moisture accumulation and resistant to precipitation.

These conclusions were strongly confirmed when we considered a subset of the summer climatology associated with the 20 most intense precipitation events to occur between 1980–2007 for our three regions of interest. This provides us with generalised pathways for extreme events, which indicate that what we observed in the case studies was consistent with the mechanisms for precipitation extremes, in general, providing a basis for anticipating the weather regimes which may lead to such events.

The third case study considered was the *K* flood of August 2017. In contrast to the previous two case studies, the characteristic pathways were rather similar to the climatology; however, a significant number of parcels were observed to originate from the mid-latitudes. The generalised pathways of the 20 strongest extreme precipitation events to hit this region between 1980–2007 suggest that the proportion of parcels originating from the mid-latitudes was greater than for the climatology. This, again, indicates the importance of the coupling between mid-latitudinal and monsoon weather systems. However, while, for the western area of the Hindu-Kush-Himalayan region (*HKH*), an increased influence of the monsoon is crucial, the opposite can be said for the eastern portion of the *HKH*, where anomalous increases in the influence of parcels originating from the mid-latitudes is associated with extreme precipitation. In all cases, however, the common feature present across the three case studies was the proximity of upper-level troughs to the target regions.

As much of the moisture gains occurred over the Indus basin for all studies here, the accurate representation of irrigation in climate models is important to correctly represent both local [7,8] and global [6] water cycles. These findings demonstrate the impact of teleconnections on regional and seasonal changes of moisture sources across the *HKH* and provide insight into the related mechanisms. In fact, the coupled role played by the mid-latitudinal and sub-tropical weather systems in delivering extreme events makes assessment of the ultimate fate of extreme events difficult to gauge in a warming climate. On one hand, continued warming leads to a decreases in the equator-to-pole temperature gradient through latent heat transport, which decreases baroclinicity and the potential for development of deep troughs, which we suggest here to be one key factor. Conversely, warming leads to an increase in available water to supply extreme events in locations where orographic lifting will always be present. How this will impact the pathways of water transport in orographically complex regions at the borderline between the mid-latitudes and tropics, such as those analysed here, is hard to anticipate, and should be investigated in detail using improved climate models.

In this paper, we have investigated hydroclimatology and extreme events in the Indian subcontinent; however, the strategies followed here could be replicated for many other regions and case studies. Furthermore, this is a useful tool in testing the skill of climate models to be able to replicate moisture pathways in regions during problematic periods, where models are known to perform poorly, such as in the case here of the *HKH* region during the monsoon. This can help validate whether climate models which appear to perform coherently with observations are doing so for the right reasons and help to diagnose why poorly performing models are in fact doing so; see, for example, the offline implementation of FLEXPART for studying transport in the CMIP5-version of the Norwegian Earth System Model [60]. This may, thus, prove to be a useful tool for the improvement of the performance of models over some key regions of interest.

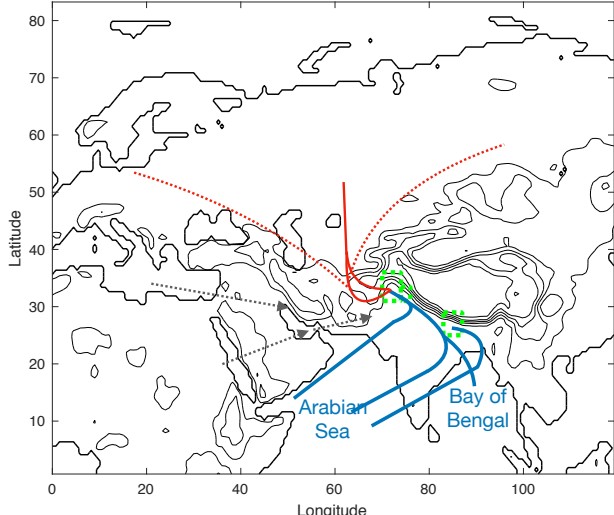

**Figure 17.** Schematic view of the parcel pathways during the monsoon season. Red dashed lines are the bounds for the main climatology for parcel back-trajectories originating from Eastern Europe and Russia. Gray arrows are for the winter. Solid blue are for the extreme events discussed.

**Author Contributions:** Conceptualization by V.L.; R.B. designed the research, processed the meteorological data, modified the numerical model, and wrote the manuscript; R.B. and V.L. contributed scientific discussion; V.L. supervised this research; R.B. and V.L. contributed to reviewing the manuscript.

**Funding:** The Authors are funded by the BITMAP-WATER (BMBF/Belmont Forum) project (Grant Number: 01LP1608A).

**Acknowledgments:** The authors wish to thank H. Fowler, N. Forsythe, K. Hunt, and A. Turner for many stimulating conversations, as well as S. Hagemann for inputs on an earlier version of the manuscript. Data and software tools used in this research work are available on request from the corresponding author.

**Conflicts of Interest:** The authors declare no conflict of interest.

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
