# Peer review of "Water Pathways for the Hindu-Kush-Himalaya and an Analysis of Three Flood Events"

_atmosphere, doi:10.3390/atmos10090489_

Round 1

Reviewer 1 Report

Review about the manuscript "Water pathways for the Hindu-Kush-Himalaya and Analysis of Three Flood Events"

This manuscript presents an intersting study about the sources and sinks of moisture for the Hindu-Kush-Himalaya region. The authors use the Flexpart model to account for the transport of moisture to the region of interest. The authors find that transport of moist air in the low levels is an important factor for moisture over the region of interest, including the contribution of tropical moisture in the climatology and low-level convergence during extreme weather events. The authors also describe the differences in sources between summer and winter, describing the role of tropical moisture and the transport associated to mid-latitude weather systems. After some revisions this manuscript is worth of publication and an interesting contribution to the literature on atmospheric moisture transport. Therefore I recommend publication of this manuscript after Minor Revisions. Below you will find some comments that I think can help to improve the current version of the manuscript.

General Comments:
=================

1. This work fits into a general area of study about water vapor transport and sources/sinks of atmospheric moisture. In this sense the authors can include more references to this kind of studies, starting with some other studies using Flexpart (see e.g. review by Gimeno et al. 2012) for other parts of the world and contrasting with studies that use Eulerian methods (e.g. Newmann et al. 2012). In addition it would be interesting to see some discussion about the relative roles of advection of moisture vs. recycling and convergence associated to deformation of streamlines (i.e. convergence not only associated to long-range transport). In this respect see e.g. Dacre et al. 2015.

2. The authors refer to the limitations of current Earth System and Climate models regarding the representation of orographic details, and their corresponding effects on atmospheric flow and precipitation. In this work the authors use ERA-Interim, and the transport of moisture with Flexpart uses wind fields at 1°, not very far from the resolution of some climate models. It would be interesting that the authors include some discussion about the potential impacts/limitations of their study associated with the ERA-Interim 1° input data.

3. Since Flexpart is a well known transport model, it would be interesting to see what are the novel contributions of this manuscript regarding the use and analysis of Flexpart output (e.g. altitude and density of air parcels?). Please state these contributions clearly in the Abstract and Introduction.

4. In general the manuscript is well organized. However the English needs revisions in several parts of the manuscript.

Specific Comments:
=================

L002. The moisture transport model used is Flexpart, which is already known in the literature. It is important to state in the Abstract that Flexpart is the model used in this study.

L031. The reader is refered to Figure 1 for an illustration of the region of interest. However, the current Figure 1 is an schematic of the calculation of sources and sinks. Please add a new Figure 1 for the region of interest, including the important topographic (mountain ranges) and hydrologic (river basins) features.

L038. Change "were highlighted" by "was highlighted", as it refers to "An unexpected consequence".

L038-040. Please re-word these lines to be more clear. For example, to what kind of changes are you refering to in line 38?.

L041-085. Reduce the part about Earth-system models and glaciers in the introduction, lines 41-85. This is important motivation for the current study, but it is not related to the main question/methods of the study. Maybe the authors can reduce lines 41-85 to one paragraph that still shows the importance of Earth-system models (resolution) and glaciers.

L125. Please assign a number to Nieto et al. 2006.

L133. Delete question mark in "[19,34?-36]"

L137. Change question mark by corresponding reference.

L139. Please cite some examples of the Eulerian analyses you are refering to.

L205-206. Please include a reference about the 80% threshold in the model for ERA-Interim.

L210. You define q in equation 1, but not Delta_q.

L224. You write "provides use". I think you meant "provides us".

L244. For Figure 2, can you please add a vector field or streamlines that indicate the direction of the flow? This can provide the reader with an idea of the flow in order to better identify the regions upstream and downstream for a particular target region. You can plot winds for the composites D-2, D-6 and D-10, or maybe just the average for the season. Maybe you can also add information about the winds in Figures 3 and 4, and 5, 6 and 7.

L254-255. It reads awkward. Please re-word this sentence.

L255. In the caption of Figure 4 you have some information about the tropical or midlatitude origin of air parcels. Please also include and discuss this result in the main text.

L279-288. These lines are almost identical to lines 262-269.

L381. Please define WD.

L391. Since you are talking about Weather Events, it would be more appropriate to talk about intra-seasonal than inter-seasonal variations, or simply synoptic time scales.

L399. Fig. 8, I suggest to remove the contour labels (or at least round to integers and reduce the number of contour labels) to make the plots more clear. Similarly for Figs. 9-11.

L400. Change "in the case" by "In the case". In addition please include in this part that Fig. 10 if for U on 16/06/2013 and Fig. 11 for K on 13/08/2017.

L609. There is no mention in the text of Figure 15. Please include a description and discussion in the main text about Figure 15.

References:
===========

Dacre, H. F., Clark, P. A., Martinez-Alvarado, O., Stringer, M. A., & Lavers, D. A. (2015). How do atmospheric rivers form?. Bulletin of the American Meteorological Society, 96(8), 1243-1255.

Gimeno, L., Stohl, A., Trigo, R. M., Dominguez, F., Yoshimura, K., Yu, L., ... & Nieto, R. (2012). Oceanic and terrestrial sources of continental precipitation. Reviews of Geophysics, 50(4).

Newman, M., Kiladis, G. N., Weickmann, K. M., Ralph, F. M., & Sardeshmukh, P. D. (2012). Relative contributions of synoptic and low-frequency eddies to time-mean atmospheric moisture transport, including the role of atmospheric rivers. Journal of Climate, 25(21), 7341-7361.

Author Response

Review about the manuscript "Water pathways for the Hindu-Kush-Himalaya and Analysis of Three Flood Events"

This manuscript presents an intersting study about the sources and sinks of moisture for the Hindu-Kush-Himalaya region. The authors use the Flexpart model to account for the transport of moisture to the region of interest. The authors find that transport of moist air in the low levels is an important factor for moisture over the region of interest, including the contribution of tropical moisture in the climatology and low-level convergence during extreme weather events. The authors also describe the differences in sources between summer and winter, describing the role of tropical moisture and the transport associated to mid-latitude weather systems. After some revisions this manuscript is worth of publication and an interesting contribution to the literature on atmospheric moisture transport. Therefore I recommend publication of this manuscript after Minor Revisions. Below you will find some comments that I think can help to improve the current version of the manuscript.

We thank the reviewer for his/her nice words and encouragement. We have tried to improve the manuscript taking into account the indicated criticisms and suggestions.

General Comments:
=================

This work fits into a general area of study about water vapor transport and sources/sinks of atmospheric moisture. In this sense the authors can include more references to this kind of studies, starting with some other studies using Flexpart (see e.g. review by Gimeno et al. 2012) for other parts of the world and contrasting with studies that use Eulerian methods (e.g. Newmann et al. 2012).

We thank the reviewer for suggesting these extremely relevant references. They have now been included.

In addition it would be interesting to see some discussion about the relative roles of advection of moisture vs. recycling and convergence associated to deformation of streamlines (i.e. convergence not only associated to long-range transport). In this respect see e.g. Dacre et al. 2015.

We feel that this issue a bit beyond the scopes of the present study.

The authors refer to the limitations of current Earth System and Climate models regarding the representation of orographic details, and their corresponding effects on atmospheric flow and precipitation. In this work the authors use ERA-Interim, and the transport of moisture with Flexpart uses wind fields at 1°, not very far from the resolution of some climate models. It would be interesting that the authors include some discussion about the potential impacts/limitations of their study associated with the ERA-Interim 1° input data.

We have added reference to implementations of FLEXPART for limited area models and made a comment in the direction suggested by the reviewer; in our case, since we are looking at very large-scale transport phenomena, the classic version is to be preferred.

Since Flexpart is a well known transport model, it would be interesting to see what are the novel contributions of this manuscript regarding the use and analysis of Flexpart output (e.g. altitude and density of air parcels?). Please state these contributions clearly in the Abstract and Introduction.

Since the scope of the paper is to look at the atmospheric processes rather than describing the Flexpart model customization we have introduced here, we have kept the description of the model in Section 2. Nonetheless, we have made explicit reference to the fact that we are not using the standard version in the abstract and in the Introduction.

In general the manuscript is well organized. However the English needs revisions in several parts of the manuscript.

Specific Comments:
=================

L002. The moisture transport model used is Flexpart, which is already known in the literature. It is important to state in the Abstract that Flexpart is the model used in this study.

Done

L031. The reader is refered to Figure 1 for an illustration of the region of interest. However, the current Figure 1 is an schematic of the calculation of sources and sinks. Please add a new Figure 1 for the region of interest, including the important topographic (mountain ranges) and hydrologic (river basins) features.

Done.

L038. Change "were highlighted" by "was highlighted", as it refers to "An unexpected consequence".

Done

L038-040. Please re-word these lines to be more clear. For example, to what kind of changes are you refering to in line 38?.

Done

L041-085. Reduce the part about Earth-system models and glaciers in the introduction, lines 41-85. This is important motivation for the current study, but it is not related to the main question/methods of the study. Maybe the authors can reduce lines 41-85 to one paragraph that still shows the importance of Earth-system models (resolution) and glaciers.

We appreciate the reviewer’s comment; nonetheless, we prefer to keep the detail as it is now because we want to provide a broader context to our study.

L125. Please assign a number to Nieto et al. 2006.

Done

L133. Delete question mark in "[19,34?-36]"

Done

L137. Change question mark by corresponding reference.

Done

L139. Please cite some examples of the Eulerian analyses you are refering to.

Done.

L205-206. Please include a reference about the 80% threshold in the model for ERA-Interim.

Done.

L210. You define q in equation 1, but not Delta_q.

Now it has been clarified.

L224. You write "provides use". I think you meant "provides us".

Done

L244. For Figure 2, can you please add a vector field or streamlines that indicate the direction of the flow? This can provide the reader with an idea of the flow in order to better identify the regions upstream and downstream for a particular target region. You can plot winds for the composites D-2, D-6 and D-10, or maybe just the average for the season. Maybe you can also add information about the winds in Figures 3 and 4, and 5, 6 and 7.

We have added the climatology on winds for summer and winter seasons in Figs 3 and 6, respectively

L254-255. It reads awkward. Please re-word this sentence.

Done

L255. In the caption of Figure 4 you have some information about the tropical or midlatitude origin of air parcels. Please also include and discuss this result in the main text.

We have now added discussion on this in sections 3.1.1, 3.1.2 and 3.1.3.

L279-288. These lines are almost identical to lines 262-269.

Removed

L381. Please define WD.

Done

L391. Since you are talking about Weather Events, it would be more appropriate to talk about intra-seasonal than inter-seasonal variations, or simply synoptic time scales.

Done

L399. Fig. 8, I suggest to remove the contour labels (or at least round to integers and reduce the number of contour labels) to make the plots more clear. Similarly for Figs. 9-11.

Done

L400. Change "in the case" by "In the case". In addition please include in this part that Fig. 10 if for U on 16/06/2013 and Fig. 11 for K on 13/08/2017.

Done

L609. There is no mention in the text of Figure 15. Please include a description and discussion in the main text about Figure 15.

Done.

Reviewer 2 Report

The study appears quite satisfactory in terms of number of data and analyzes conducted. The range is quite large and more local information may not completely confirm the results of the work. This, however, represents a basic compendium for other research. The results are solidly supported, I would improve only the visibility of some figures and enrich the bibliographic support to the contents.
It would be appropriate to cite in the introduction some studies in other research contexts to see if the influence of glacialized or deglacialized areas and the occurrence of extreme events, even non-monsoon, could be a nice scientific parallelism. Some works in the Italian context are suggested below, which could allow food for thought.

PARANUNZIO R., CHIARLE M., LAIO F., NIGRELLI G., TURCONI L., LUINO F. (2018) - New insights in the relation between climate and slope failures at high-elevation sites, Theoretical and Applied Climatology (Internet, https://link.springer.com/article/10.1007%2Fs00704-018-2673-4

TURCONI L., SUNIL DE K., TROPEANO D., SAVIO G. (2010) – “Slope failure and related processes in the Mt. Rocciamelone area (Cenischia valley, Western Italian Alps), Geomorphology 114, 115-128

Author Response

The study appears quite satisfactory in terms of number of data and analyzes conducted. The range is quite large and more local information may not completely confirm the results of the work. This, however, represents a basic compendium for other research. The results are solidly supported, I would improve only the visibility of some figures and enrich the bibliographic support to the contents.

We thank the reviewer for his/her nice words and appreciation of our work.

It would be appropriate to cite in the introduction some studies in other research contexts to see if the influence of glacialized or deglacialized areas and the occurrence of extreme events, even non-monsoon, could be a nice scientific parallelism. Some works in the Italian context are suggested below, which could allow food for thought.

PARANUNZIO R., CHIARLE M., LAIO F., NIGRELLI G., TURCONI L., LUINO F. (2018) - New insights in the relation between climate and slope failures at high-elevation sites, Theoretical and Applied Climatology (Internet, https://link.springer.com/article/10.1007%2Fs00704-018-2673-4

TURCONI L., SUNIL DE K., TROPEANO D., SAVIO G. (2010) – “Slope failure and related processes in the Mt. Rocciamelone area (Cenischia valley, Western Italian Alps), Geomorphology 114, 115-128

Thanks for the useful suggestion. We were not aware of this literature. We made reerence to the impact of climate change on landslides in the introduction.